

# A global Budyko model to partition evaporation into interception and transpiration

**Ameneh Mianabadi[1,2], Miriam Coenders–Gerrits[2*], Pooya Shirazi[1], Bijan Ghahraman[1], Amin Alizadeh[1]**

1- **Ferdowsi University of Mashhad, Mashhad, Iran**
2- **Delft University of Technology, Delft, The Netherlands**

*Corresponding author*

## Abstract

Evaporation is a very important flux in the hydrological cycle and links the water and energy balance of a catchment. The Budyko framework is often used to provide a first order estimate of evaporation, since it is a simple model where only rainfall and potential evaporation is required as input. Many researchers have tried to improve the Budyko framework by including more physics and catchment characteristics into the original equation. However, this often resulted in additional parameters, which are unknown or difficult to determine. In this paper we present an improvement of the previously presented Gerrits' model ("Analytical derivation of the Budyko curve based on rainfall characteristics and a simple evaporation model" in Gerrits et al, 2009 WRR), whereby total evaporation is calculated on the basis of simple interception and transpiration thresholds in combination with measurable parameters like rainfall dynamics and storage availability from remotely sensed data sources. While Gerrits' model was investigated for 10 catchments with different climate conditions and some parameters were assumed to be constant, in this study we applied the model on the global scale and fed with remotely sensed input data. The output of the model has been compared to two complex land-surface models STEAM and GLEAM, as well as the database of Landflux-EVAL. Our results show that total evaporation estimated by Gerrits' model is in good agreement with Landflux-EVAL, STEAM and GLEAM. Results also show that Gerrits' model underestimates interception in comparison to STEAM and overestimates it in comparison to GLEAM, while for transpiration the opposite is found. Errors in interception can partly be explained by differences in the interception definition that successively introduce errors in the calculation of transpiration. Comparing to the Budyko framework, the model showed a good performance for total evaporation estimation.

**Keywords:** Budyko curves, interception, transpiration, remote sensing, evaporation




## 1 Introduction

Budyko curves are used as a first order estimate of annual evaporation as a function of annual precipitation and potential evaporation. If the available energy is sufficient to evaporate the available moisture, annual evaporation can approach annual precipitation (water-limited situation). If the available energy is not sufficient, annual evaporation can approach potential evaporation (energy-limited situation). Using the water balance and the energy balance and by applying the definition of the aridity index and Bowen ratio, the Budyko framework can be described as (Arora, 2002):

$$\frac{E_a}{P_a} = \frac{\emptyset}{1+f(\emptyset)} = F(\emptyset) \tag{1}$$

with $E_a$ annual evaporation [L/T], $P_a$ annual precipitation [L/T], $\frac{E_a}{P_a}$ the evaporation ratio [-], and $\emptyset$ the aridity index which is defined as the potential evaporation divided by annual precipitation [-]. Equation 1 is the base of all Budyko curves, which are developed by different researchers (Table 1).

The equations shown in Table 1 assume that the evaporation ratio is determined by climate only and do not take into account the effect of other controls on the water balance. Therefore, some researchers tried to incorporate more physics into the Budyko framework. For example Milly (1994, 1993) investigated the root zone storage as an important secondary control on the water balance. Choudhury (1999) used net radiation and a calibration factor in Budyko curves. Zhang et al. (2004, 2001) tried to add a plant-available water coefficient, Porporato et al. (2004) took into account the maximum storage capacity, Yang et al. (2006, 2008) incorporated a catchment parameter, and Donohue et al. (2007) tried to consider vegetation dynamics. Although the incorporation of these additional processes improved the model performance, the main difficulty with these approaches is the determination of the parameter values. In practice, they are therefore often used as calibration parameters. The model of Gerrits et al. (2009) (hereafter Gerrits' model) aimed to develop an analytical model that is physically based and only uses measurable parameters. They tested the model output (i.e., interception evaporation, transpiration, and total evaporation) on a couple of locations in the world, where the parameters could be determined, but not at the global scale due to data limitations. However, with the current developments in remotely sensed data new opportunities have arisen.

Recently, many studies (e.g., Chen et al., 2013; Donohue et al., 2010; Istanbulluoglu et al., 2012; Milly and Dunne, 2002; Wang, 2012; Zhang et al., 2008) found that soil moisture storage change is a critical component in modelling the interannual water balance. Including soil water information into the Budyko framework was often difficult, because this information is not widely available. However, Gao et al. (2014) presented a new method where the available soil water is derived from time series of rainfall and potential evaporation, plus a long-term runoff coefficient. This data can be derived locally (e.g., de Boer-Euser et al. (2016)), but can also be derived from remotely sensed data as shown by Wang-Erlandsson et al. (2016), which allows us to apply the method at the global scale and incorporate it in the Gerrits' model.

While Gerrits' model was only tested for 10 locations with different climatic conditions, the aim of this study is to test Gerrits' model at the global scale. We used remotely sensed data to estimate





parameters, which were considered constant in Gerrits' model. These parameters are the maximum
soil moisture storage by the method of Gao et al (2014) and the interception storage capacity.
These parameters are required to make a first order estimate of total evaporation, and to partition
this into interception evaporation and transpiration as well. The outcome is compared to more
complex land-surface-atmosphere models as well as to the Budyko curves of Table 1.
**Methodology**
Total evaporation ($E$) may be partitioned as follows (Shuttleworth, 1993):

$$E = E_i + E_t + E_o + E_s \qquad\qquad (2)$$

in which $E_i$ is interception evaporation, $E_t$ is transpiration, $E_o$ is evaporation from water bodies
and $E_s$ is evaporation from the soil, all with dimensions [LT$^{-1}$]. In this definition, interception is
the amount of evaporation from any wet surface including canopy, understory, forest floor, and
the top layer of the soil. Soil evaporation is defined as evaporation of the moisture in the soil that
is connected to the root zone (de Groen and Savenije, 2006) and therefore is different from
evaporation of the top layer of the soil (several millimeters of soil depth, which is here considered
as part of the interception evaporation). Hence interception evaporation is the fast feedback of
moisture to the atmosphere within a day from the rainfall event and soil evaporation is evaporation
from the soil constrained by soil moisture storage in the root zone. Like Gerrits et al. (2009), we
assume that evaporation from soil moisture is negligible (or can be combined with interception
evaporation). Evaporation from water bodies is used for inland open water, where interception
evaporation and transpiration is zero. As a result, Equation 2 becomes:

$$E = E_o \qquad\qquad \text{for water bodies} \qquad\qquad (3a)$$
$$E = E_i + E_t \qquad\qquad \text{for land surface} \qquad\qquad (3b)$$

where $E_i$ is direct feedback from short term moisture storage on vegetation, ground, and top layer,
and $E_t$ is evaporation from soil moisture storage in the root zone.
For modelling evaporation, it is important to consider that interception and transpiration have
different time scales (i.e. the stock divided by the evaporative flux) (Blyth and Harding, 2011).
With a stock of a few millimetres and the evaporative flux of a few millimetres per day,
interception has a time scale in the order of one day (Dolman and Gregory, 1992; Gerrits et al.,
2009, 2007; Savenije, 2004; Scott et al., 1995). In the case of transpiration, the stock amounts to
tens to hundreds of millimetres and the evaporative flux to a few millimetres per day (Baird and
Wilby, 1999), resulting in a time scale in the order of month(s) (Gerrits et al., 2009). In Gerrits'
model it is successively assumed that interception and transpiration can be modelled as threshold
processes at the daily and monthly time scale, respectively. Rainfall characteristics are
successively used to temporally upscale from daily to monthly, and from monthly to annual. A full
description of the derivation and assumptions can be found in Gerrits et al. (2009). Here, we only
summarize the relevant equations (Table 2) and not the complete derivation. Since we now test the
model at the global scale, we do show how we estimated the required model parameters and the
inputs used.





## 1 Interception

Gerrits' model considers evaporation from interception as a threshold process at daily time scale (Equation 4, Table 2). Daily interception ($E_{i,d}$), then, is upscaled to monthly interception ($E_{i,m}$, Equation 5, Table 2) by considering the frequency distribution of rainfall on a rain day ($\beta$-parameter) and subsequently to annual interception ($E_{i,a}$, Equation 6, Table 2) by considering the frequency distribution of rainfall in a rain month ($\kappa_m$-parameter) (see de Groen and Savenije (2006), Gerrits et al. (2009)). A rain day is defined as a day with more than 0.1 mm day$^{-1}$ of rain and a rain month is a month with more than 2 mm month$^{-1}$ of rain.

While Gerrits et al. (2009) assumed a constant interception threshold ($D_{i,d} = 5$ mm day$^{-1}$) for the studied locations, we here use a globally variable value based on remote sensing data. The interception threshold ($D_{i,d}$) is a yearly average and is either limited by the daily interception storage capacity $S_{max}$ (mm day$^{-1}$) or by the daily potential evaporation (Equation 9, Table 2) and thus not seasonally variable. We can assume this, because for most locations $S_{max}$ is smaller than $E_{p,d}$ even if we consider a daily varying potential evaporation. Additionally, $S_{max}$ (based on LAI) could also be changed seasonally, however many studies show that the storage capacity is not changing significantly between the leafed and leafless period (e.g., Leyton et al., 1967; Dolman, 1987; Rutter et al., 1975). Especially, once interception is defined in a broad sense that it includes all evaporation from the canopy, understory, forest floor, and the top layer of the soil: leaves that are dropped from the canopy remain their interception capacity as they are on the forest floor in the leafless period. Furthermore, Gerrits et al (2010) showed with a Rutter-like model that interception is more influenced by the rainfall pattern than by the storage capacity, which was also found by Miralles et al. (2010). Hence, in interception modelling, the value of the storage capacity is of minor concern, and seasonality is incorporated in the temporal rainfall patterns.

The daily interception storage capacity should be seen as the maximum interception capacity within one day, including the (partly) emptying and filling of the storage between events per day, thus $S_{max} = n \cdot C_{max}$, where $C_{max}$ [L] is the interception storage capacity of land cover. If there is only one rain event per day ($n = 1$ day$^{-1}$) (Gerrits et al., 2010), $S_{max}$ [LT$^{-1}$] equals $C_{max}$ [L], as is often found in literature. Despite proposing modifications for storms, which last more than one day (Pearce and Rowe, 1981), and multiple storms per rain day (Mulder, 1985), accounting for $n$ is rarely necessary (Miralles et al., 2010).

For $n = 1$, the interception storage capacity can be estimated from Von Hoyningen-Huene (1981), which is obtained for a series of crops based on the leaf area index (LAI) (de Jong and Jetten, 2007) (Equation 10, Table 2). Since the storage capacity of the forest floor is not directly related to LAI, it could be said that the 0.935 mm in Equation 10 is sort of the storage capacity of the forest floor. Since this equation was developed for crops, it is likely that it underestimates interception by forests with a denser understory and forest floor interception capacity.

## 37 Transpiration





Transpiration is considered as a threshold process at the monthly time scale ($E_{t,m}$ (mm month$^{-1}$),
Equation 7, Table 2) and successively is upscaled to annual transpiration ($E_{t,a}$ (mm year$^{-1}$),
Equation 8, Table 2) by considering the frequency distribution of the net monthly rainfall ($P_{n,m} =$
$P_m - E_{i,m}$) expressed with the parameter $\kappa_n$. To estimate the monthly and annual transpiration,
two parameters $A$ and $B$ are required. $A$ is the initial soil moisture or carryover value (mm month$^{-}$
$^1$) and $B$ is dimensionless and described as Equation 15, where the dimensionless $\gamma$ is obtained by
Equation 16.
Gerrits et al. (2009) assumed that the carry over value ($A$) is constant and used $A = 0, 5, 15, 20$,
mm month$^{-1}$ , depending on the location, to determine annual transpiration. Also they considered
$\gamma$ to be constant ($\gamma = 0.5$). In the current study, we determined these two parameters based on the
maximum root zone storage capacity ($S_{u,max}$). In  equation 16, $\Delta t_m = 1$ month and $S_b$ can be
assumed to be 50% to 80% of $S_{u,max}$ (de Groen, 2002; Shuttleworth, 1993). In this study we
assumed $S_b$ to be 50% of $S_{u,max}$ as this value is commonly used for many crops (Allen et al.,
1998). Furthermore, we assumed that the monthly carry over $A$ can be estimated as $bS_{u,max}$  and
in this study we assumed $b = 0.2$ which gave the best global results for all land classes. To estimate
$A$ and $\gamma$, it is important to have a reliable database of $S_{u,max}$. For this purpose, we used the global
estimation of $S_{u,max}$ from Wang-Erlandsson et al. (2016) (Fig. 1d). $S_{u,max}$ is derived by the mass
balance method using satellite based precipitation and evaporation (Wang-Erlandsson et al., 2016).
Wang-Erlandsson et al. (2016) estimated the root zone storage capacity from the maximum soil
moisture deficit, as the integral of the outgoing flux (i.e. evaporation which is sum of transpiration,
evaporation, interception, soil moisture evaporation and open water evaporation) minus the
incoming flux (i.e. precipitation and irrigation). In their study, the root zone storage capacity was
defined as the total amount of water that plants can store to survive droughts. Note that this recent
method (Gao et al., 2014) to estimate $S_{u,max}$ does not require soil information, but only uses
climatic data. It is assumed that ecosystems adjust their storage capacity to climatic demands
irrespective of the soil properties. Under wet conditions Gao's method appeared to perform better.
For (semi-)arid climates the difference between this method and soil-based methods appear to be
small (de Boer-Euser et al., 2016).
Furthermore, Gerrits et al. (2009) estimated the average monthly transpiration threshold ($D_{t,m}$) as
$\frac{E_p - E_{i,a}}{n_a}$ (where $n_a$= number of months per year), which assumes that if there is little interception,
plants can transpire at the same rate as a well-watered reference grass as calculated with the
Penman-Monteith equation (University of East Anglia Climatic Research Unit , 2014). In reality,
most plants encounter more resistance (crop resistance) than grass, hence we used Equation 17,
Table 2 (Fredlund et al., 2012) to convert potential evaporation of reference grass ($E_p$) to potential
transpiration of a certain crop depending on LAI (i.e. the transpiration threshold $D_{t,m}$ [mm month$^{-}$
$^1$]). Furthermore, similar to the daily interception threshold, we took a constant $D_{t,m}$, which can be
problematic in energy-constrained areas. But in those relatively wet areas transpiration is
underestimated in summer, but overestimated in winter, which will cancel out on the annual scale.
**Data**





For precipitation we used the AgMERRA product from AgMIP climate forcing dataset (Ruane et
al., 2015), which has a daily time scale and a spatial resolution of 0.25°×0.25° (see Fig. 1a). The
spatial coverage of AgMERRA is globally for the years 1980-2010. The AgMERRA product is
available on the NASA Goddard Institute for Space Studies website
(http://data.giss.nasa.gov/impacts/agmipcf/agmerra/).
Potential evaporation (see Fig. 1b) data (calculated by FAO-Penman–Monteith equation (Allen et
al., 1998)) were taken from Center for Environmental Data Archival website
(http://catalogue.ceda.ac.uk/uuid/4a6d071383976a5fb24b5b42e28cf28f), produced by the
Climatic Research Unit (CRU) at the University of East Anglia (University of East Anglia Climatic
Research Unit, 2014). These data are at the monthly time scale over the period 1901-2013, and has
a spatial resolution of 0.5°×0.5°. We used the data of 1980-2010 in consistent with precipitation
dataset.
LAI data (Fig. 1c) were obtained from Vegetation Remote Sensing & Climate Research
(http://sites.bu.edu/cliveg/datacodes/) (Zhu et al., 2013). The spatial resolution of the data sets is
1/12 degree, with 15-day composites (2 per month) for the period July 1981 to December 2011.
The data of $S_{u,max}$ (Fig. 1d) is prepared data by Wang-Erlandsson et al. (2016) and is based on
the satellite based precipitation and evaporation with 0.5°×0.5° resolution over the period 2003-
2013. They used the USGS Climate Hazards Group InfraRed Precipitation with Stations (CHIRPS)
precipitation data at 0.05° (Funk et al., 2014) and the ensemble mean of three datasets of
evaporation including CSIRO MODIS Reflectance Scaling EvapoTranspiration (CMRSET) at
0.05° (Guerschman et al., 2009), the Operational Simplified Surface Energy Balance (SSEBop) at
30″ (Senay et al., 2013) and MODIS evapotranspiration (MOD16) at 0.05° (Mu et al., 2011). They
calculated potential evaporation using Penman-Monteith equation (Monteith, 1965).
**Model comparison and evaluation**
The model performance was evaluated by comparing our results at the global scale to global
evaporation estimates from other studies. Most available products only provide total evaporation
estimates and do not distinguish between interception and transpiration. Therefore, we chose to
compare our interception and transpiration results to two land surface models: The Global Land
Evaporation Amsterdam Model (GLEAM) (v3.0a) database (Martens et al., 2017; Miralles et al.,
2011a) and Simple Terrestrial Evaporation to Atmosphere Model (STEAM) (Wang-Erlandsson et
al., 2014, Wang-Erlandsson et al., 2016). GLEAM estimates different fluxes of evaporation
including transpiration, interception, bare soil evaporation, snow sublimation and open water
evaporation. STEAM, on the other hand, estimates the different components of evaporation
including transpiration, vegetation interception, floor interception, soil moisture evaporation, and
open water evaporation. Thus for the comparison of interception we used the sum of canopy and
floor interception and soil evaporation from STEAM and canopy interception and bare soil
evaporation from GLEAM. Furthermore, STEAM includes an irrigation module (Wang-
Erlandsson et al., 2014), while Miralles et al. (2011) mentioned that they did not include irrigation
in GLEAM, but the assimilation of the soil moisture from satellite would account for it as soil
moisture adjusted to seasonal dynamics of any region. The total evaporation was also compared to
LandFlux-EVAL products (Mueller et al., 2013). GLEAM database (www.gleam.eu) is available





for 1980-2014 with a resolution of 0.25°×0.25° and STEAM model was performed for 2003-2013
with a resolution of 1.5°×1.5°. LandFlux-EVAL data (https://data.iac.ethz.ch/landflux/) is
available for 1989-2005. We compared Gerrits' model to other products based on the land cover
to judge the performance of the model for different types of land cover. The global land cover map
(Channan et al., 2014; Friedl et al., 2010) was obtained from http://glcf.umd.edu/data/lc/. Lastly,
we also compared our results to the Budyko curves of Schreiber, O'ldekop, Pike and Budyko
(Table 1). We used root mean square error (*RMSE*) (Eq. 20), mean bias error (*MBE*) (Eq. 21) and
relative error (*RE*) (Eq. 22) to evaluate the results.

$$\text{RMSE} = \sqrt{\frac{\sum_{i=1}^{n}(x_{iG} - x_{iM})^2}{n}} \tag{20}$$

$$\text{MBE} = \frac{\sum_{i=1}^{n}(x_{iG} - x_{iM})}{n} \tag{21}$$

$$\text{RE} = \frac{\bar{x}_G - \bar{x}_M}{\bar{x}_G} \times 100 \tag{22}$$

In these equations, $x_{iM}$ is evaporation of the benchmark models to which Gerrits' model is
compared for pixel $i$, $x_{iG}$ is evaporation from Gerrits' model for pixel $i$, $\bar{x}_G$ is the average
evaporation of Gerrits' model, $\bar{x}_M$ is the average evaporation of the benchmark models and $n$ is
the number of pixels of the evaporation map. Negative MBE and RE show the Gerrits' model
underestimates evaporation and vice versa. As the spatial resolution of the products is different,
we regridded all the products to the coarsest resolution (1.5°×1.5°) for the comparison.
Furthermore, the comparisons were shown for each land cover using the Taylor diagram (Taylor,
2001). This diagram can provide a concise statistical summary of how the models are comparable
to the reference data (observation or given model) in terms of their correlation, RMSE, and the
ratio of their variances. In this paper, the reference data is Gerrits' model. Since the different
models for different land cover types have been used in this study, which have different numerical
values, the results are normalized by the reference data. It should be Noted that the standard
deviation of the reference data is normalized by itself and, therefore, it is plotted at unit distance
from the origin along the horizontal axis (Taylor, 2001). According to Taylor diagram, when the
points are close to reference data (Ref in Figures 3, 5, 7 and 9), it shows that the RMSE is less and
the correlation is higher and therefore, the models are in a more reasonable agreement.
**Results and discussion**
**Total evaporation comparison**
Figure 2 shows the mean annual evaporation from Gerrits' model, Landflux-EVAL, STEAM and
GLEAM data sets. In general, the spatial distribution of Gerrits' simulated evaporation is similar
to that of the benchmark models. Figure 2a demonstrates that, as expected, the highest annual
evaporation, which is the sum of interception evaporation and transpiration, occurs in tropical
evergreen broadleaf forests and the lowest rate occurs in the barren and sparsely vegetated desert
regions. Total evaporation varies between almost zero in arid regions to more than 1500 mm year-
[1] in the tropics.



As can be seen in Figure 2 there exist also large differences between STEAM, GLEAM and
Landflux-EVAL. Different products of precipitation (and other global data bases) applied for the
models is likely the reason. For example, the sensitivity of the model to the number of rain days
and rain months especially for the higher rate of precipitation (Gerrits et al., 2009) can be a
probable reason for poor performance of a model especially for evergreen forests with the highest
amount of precipitation.
Mean annual evaporation contributions per land cover type from Gerrits' model and other products
as well as RMSE, MBE and RE are shown in Table 3. Globally, mean annual evaporation
estimated (for the overlapped pixels with 1.5°×1.5° resolution) by Gerrits' model, Landflux-
EVAL, STEAM and GLEAM are 443, 469, 475 and 462 mm year$^{-1}$, respectively. Our results are
comparable to those of Haddeland et al. (2011), where the simulated global terrestrial evaporation
ranges between 415 and 586 mm year$^{-1}$ for the period 1985–1999. Generally, Gerrits' model
overestimates evaporation for most land cover types in comparison to Landflux-EVAL and
GLEAM, and underestimates in comparison to STEAM (see also MBE and RE). Since the number
of pixels covered by each land use is different, RMSE, MBE and RE cannot be comparable
between land cover types. RMSE, MBE and RE for each land cover type show that, generally,
Gerrits' model is in a better agreement with Landflux and GLEAM than STEAM. The Taylor
diagram for total evaporation estimated by Gerrits' model in comparison to Landflux-EVAL,
STEAM and GLEAM for all data (No. 1 in Fig. 3) and for each land cover type (No.2 to No.11 in
Fig. 3) also indicates that Gerrits' model has a better agreement with Landflux-EVAL and GLEAM
than STEAM model, especially for Evergreen broadleaf forest, Shrublands, Savannas and
Croplands (see also Table 3).
**Annual interception comparison**
While Wang-Erlandsson et al. (2014; 2016) estimated canopy interception, floor interception and
soil evaporation separately, in the current study we assumed that these three components of
evaporation can be lumped as interception evaporation. Figure 4 shows the mean annual
evaporation from interception at the global scale estimated by Gerrits' model, STEAM and
GLEAM. In this figure, interception from STEAM is calculated by the sum of canopy interception,
floor interception and soil evaporation. Furthermore, interception from GLEAM is calculated as
the sum of canopy interception and bare soil evaporation (GLEAM does not estimate floor
interception). In general, the spatial distribution of Gerrits' simulated interception is partly similar
to that of STEAM and GLEAM. In the tropics, with high amounts of annual precipitation and high
storage capacity due to the dense vegetation (evergreen broadleaf forests and savannas), annual
interception shows the highest values. Table 4 shows the average of interception, RMSE, MBE
and RE per land cover type. This table indicates that Gerrits' model underestimates interception in
comparison to STEAM for all land cover types. Table 4 also shows that, in comparison to GLEAM,
Gerrits' model overestimates interception for all land cover types, because in GLEAM floor
interception has not been taken into account. Figure 5 also shows that Gerrits' model is in better
agreement with STEAM (especially for Grasslands and Mixed forest) than GLEAM. The reason
for an underestimated interception in comparison to STEAM could be the role of the understory.
LAI does not account for understory, therefore maybe $S_{max}$ should be larger than modeled with
Equation 10. However, there is almost no data available to estimate the interception storage
capacity of the forest floor at the global scale.





We also compared our interception ratio $E_i/E$ (Fig. 10) with some studies that looked after
evaporation partitioning. Wang-Erlandsson defined interception in a slightly different way, hence
we compared our calculated $E_i/E$ with the sum of soil moisture evaporation ratio, vegetation
interception ratio and floor interception ratio which are presented in Fig. 5.b, 5.c and 5.d in Wang-
Erlandsson et al. (2014), respectively. While the results of Wang-Erlandsson et al. (2014) showed
that vegetation interception in arid regions with no vegetation cover is zero, soil moisture and floor
interception show a considerable percentage of total evaporation. Our results also show that $\frac{E_i}{E}$ in
arid regions is close to 100%. Therefore, the interception ratio in this study is in a reasonable
agreement with the results of Wang-Erlandsson et al. (2014). It is also comparable to the sum of
bare soil evaporation and canopy interception from GLEAM (Martens et al., 2017).
**Annual transpiration comparison**
Figure 6 illustrates the mean annual transpiration estimated by Gerrits' model, STEAM and
GLEAM. The spatial distribution is similar to the results of STEAM and GLEAM. Mean annual
transpiration varies between zero mm year$^{-1}$ for arid areas in the north of Africa (Sahara) to more
than 1000 mm year$^{-1}$ in the tropics in South America. The results show that the highest annual
transpiration occurrs in Evergreen broadleaf forests with the highest amount of precipitation and
dense vegetation (see also Table 5). Figure 6c shows that GLEAM, in comparison to Gerrits'
model, overestimates the transpiration in some regions especially in the tropics in South America
and Central Africa. Figure 6b also shows that STEAM is different from Gerrits' model over some
regions like India, western China and North America as well as in the tropics. Table 5 (MBE and
RE) also indicates that Gerrits' model underestimates transpiration in comparison to GLEAM and
overestimates in comparison to STEAM. The Taylor diagram (Fig. 7) shows global annual
transpiration of Gerrits' model is closer to that of GLEAM than STEAM, representing that the
Gerrits' model is in a more reasonable agreement with GLEAM for transpiration estimation.
Similar to the interception ratio, we also compared our transpiration ratio $E_t/E$ (Fig 10), and found
that the results are in a reasonable agreement with STEAM (See Fig. 5.a, Wang-Erlandsson et al.
(2014)) and GLEAM (See Fig. 9.e, Martens et al. (2017)). Global transpiration ratio estimated by
Gerrits' model is 71% which is comparable to the ratio estimated by other studies (e.g. 80%
(Miralles et al., 2011b), 69% (Sutanto, 2015),65% (Good et al., 2015), 62% (Maxwell and Condon,
2016), 62% (Lian et al., 2018), 61% (Schlesinger and Jasechko, 2014) 60% (Coenders-Gerrits et
al., 2014), 57% (Wei et al., 2017)), 52% (Choudhury and Digirolamo, 1998), 48% (Dirmeyer et
al., 2006) and 41% (Lawrence et al., 2007). The spatial pattern of transpiration ratio is a reasonable
agreement with those of Wei et al. (2017) and Schlesinger and Jasechko (2014).
**Budyko framework**
Figure 8 shows the mean annual evaporation derived from four non-parametric Budyko curves
(Table 1) including Schreiber (1904), Ol'dekop (1911), Pike (1964) and Budyko (1974). The
global mean annual evaporation estimated by Pike and Budyko are close (445 and 439 mm year$^{-1}$,
respectively). Schreiber underestimates the mean annual evaporation in comparison to Ol'dekop,
Pike and Budyko, especially in regions with a higher rate of evaporation. Table 6 shows the mean
annual evaporation estimated by these four curves per land cover type in comparison to Gerrits'
model as well as RMSE, MBE and RE. The results show that mean annual evaporation of Gerrits'





model for forests is closer to that of Ol'dekop and for the other land classes it is closer to that of
Budyko. Global mean annual evaporation is close to Pike where RE is almost zero. Taylor diagram
(Fig. 9) shows that, in comparison to the Budyko curves, Gerrits' model performs well for all land
cover types except for Evergreen broadleaf and Deciduous needleleaf forest. Evergreen broadleaf
forest shows a significant overestimation of evaporation by Gerrits' model in comparison to
Budyko curves. One of the reasons for these differences can be the used precipitation product as
Gerrits et al. (2009) mentioned that the number of rain months per year, is the most sensitive
parameter. Furthermore, as mentioned before in Section "Annual interception comparison", the
role of the understory, which has not been taken into account in $S_{max}$ equation, can be a source of
error for the poor interception performance (and therefore total evaporation) in forests.

**Conclusion**

In the current study we improved and applied a simple evaporation model proposed by Gerrits et
al. (2009) at the global scale. Instead of locally calibrated model parameters we now only used
parameters derived from remotely sensed data. Furthermore, we implemented in the Gerrits' model
a new definition of the root zone storage capacity from Gao et al (2014).
Comparing our results for total evaporation to Landflux-EVAL estimates show that Gerrits' model
is in good agreement with Landflux-EVAL. The highest mean annual evaporation rates are found
in evergreen broadleaf forests (1367 mm year$^{-1}$), deciduous broadleaf forests (796 mm year$^{-1}$) and
savannas (695 mm year$^{-1}$) and the lowest ones are found in shrublands (203 mm year$^{-1}$) and
grasslands (275 mm year$^{-1}$). Generally, Gerrits' model overestimates in comparison to Landflux-
EVAL and GLEAM, and underestimates in comparison to STEAM.
Gerrits' model underestimates interception in comparison to STEAM for all land covers. On the
other hand, the model overestimates interception in comparison to GLEAM, since GLEAM does
not include floor interception. Although we tried to correct for the different definitions of
interception, the results may be biased. The relatively worse performance in forests ecosystems
could be explained by the effect of understory. This is not taken into account in Gerrits' model,
while the understory can also intercept water. We could say that the constant value of 0.935 mm
in Equation 10 reflects the forest floor interception storage capacity, but since this number was
derived for crops, it is likely an underestimation. Therefore, better estimation of $S_{max}$ to account
for forest floor interception is recommended.
Estimated transpiration by Gerrits' model is in reasonable agreement with GLEAM and STEAM.
Gerrits' model underestimates transpiration in comparison to GLEAM (RE=-4%) and
overestimates in comparison to STEAM (RE=+12%). The scatter plots showed that, in comparison
to GLEAM and STEAM, Gerrits' model performs well for all land cover types. Also the
transpiration ratio corresponded well in comparison to those of GLEAM and STEAM. The results
also showed that the global transpiration ratio estimated by Gerrits' model (71%) is approximately
comparable to the other studies.
Comparing Gerrits' model to some Budyko curves, shows that the model performed well, but in
areas with few number of rain months, evaporation is not close to the Budyko curves of Schreiber,





Ol'dekop, Pike and Budyko. This is likely caused by the fact that Gerrits' model is rather sensitive to the number of rain days and rain months.

Comparing all products, we found that, in general, there are large differences between STEAM, GLEAM and Landflux-EVAL. The most convincing reason for this discrepancy lies in the different products for precipitation (and different global data sets), which have been used for the different models. The Gerrits' model is sensitive to the number of rain days and months especially for the higher rates of precipitation. Therefore, for evergreen forest with the highest amount of precipitation, this can be a probable reason for discrepancies.

Generally, it should be mentioned that the underlying reasoning of the Gerrits' model is to recognize the characteristic time scales of the different evaporation processes (i.e. interception daily and transpiration monthly). In Gerrits et al. (2009) (and in the current paper as well), this has been done by taking yearly averages for the interception ($D_{i,d}$, mm day$^{-1}$) and transpiration threshold ($D_{t,m}$, mm month$^{-1}$) in combination with the temporal distribution functions for daily and monthly (net) rainfall. Hence, the seasonality is incorporated in the temporal rainfall patterns, and not in the evaporation thresholds. This is a limitation of the currently used approach and could be the focus of a new study by investigating how seasonal fluctuating thresholds (based on LAI and/or a simple cosine function) would affect the results. This could be a significant methodological improvement of the Gerrits' model, but will have mathematical implications on the analytical model derivation. It will improve the monthly evaporation estimates, but we expect that the consequences at the annual time scale (which is the focus of the current paper) will be less severe. The strength of the Gerrits' model is that, in comparison to other models, it is a very simple and in spite of its simplicity, the Gerrits' model performs quite well.

## Acknowledgment

This research was partly funded by NWO Earth and Life Sciences (ALW), veni-project 863.15.022, the Netherlands. Furthermore, we would like to thank Iran's Ministry of Science, Research and Technology for supporting this research and the mobility fellowship. We also would like to thank Jie Zhou, Lan Wang-Erlandsson, Kamran Davary, Shervan Gharari and Hubert Savenije for their kind helps and comments.

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





1    **Table 1-** Budyko equations developed by different researchers.

| Equation | Reference |
|---|---|
| $\dfrac{E_a}{P_a} = 1 - \exp(-\emptyset)$ | Schreiber [1904] |
| $\dfrac{E_a}{P_a} = \emptyset \tanh(\dfrac{1}{\emptyset})$ | Ol'dekop [1911] |
| $\dfrac{E_a}{P_a} = \dfrac{1}{\sqrt{0.9 + (\dfrac{1}{\emptyset})^2}}$ | Turc [1954] |
| $\dfrac{E_a}{P_a} = \dfrac{1}{\sqrt{1 + (\dfrac{1}{\emptyset})^2}}$ | Pike [1964] |
| $\dfrac{E_a}{P_a} = [\emptyset \tanh\left(\dfrac{1}{\emptyset}\right)(1 - \exp(-\emptyset))]^{1/2}$ | Budyko [1974] |





**Table 2-** Summary of the interception and transpiration equations of Gerrits' model (Gerrits et al., 2009)

| Equation | Equation number | Description |
|---|---|---|
| $E_{i,d} = \min(D_{i,d}, P_d)$ | (4) | $E_{i,d}$: daily interception (mm day⁻¹), $P_d$: daily precipitation (mm day⁻¹), $D_{i,d}$: the daily interception threshold (mm day⁻¹) |
| $E_{i,m} = P_m\left(1 - \exp(-\phi_{i,m})\right)$ | (5) | $E_{i,m}$: monthly interception (mm month⁻¹), $P_m$: monthly rainfall (mm month⁻¹), $\phi_{i,m}$: a sort of aridity index for interception at monthly scale |
| $E_{i,a} = P_a\left(1 - 2\phi_{i,a}K_0(2\sqrt{\phi_{i,a}}) - 2\sqrt{\phi_{i,a}}K_1(2\sqrt{\phi_{i,a}})\right)$ | (6) | $E_{i,a}$: annual interception (mm year⁻¹), $P_a$: annual rainfall (mm year⁻¹), $\phi_{i,a}$: a sort of aridity index for interception at annual scale, $K_0$ and $K_1$: the Bessel function of the first and second order, respectively |
| $E_{t,m} = \min\left(A + B(P_m - E_{i,m}), D_{t,m}\right)$ | (7) | $E_{t,m}$: monthly transpiration (mm month⁻¹), A: carry-over parameter (mm month⁻¹), $D_{t,m}$: the transpiration threshold (mm month⁻¹), B: slope of relation between monthly effective rainfall and monthly transpiration |
| $E_{t,a} = 2BP_a\left(\phi_{i,a}K_0(2\sqrt{\phi_{i,a}}) + \sqrt{\phi_{i,a}}K_1(2\sqrt{\phi_{i,a}})\right)$ $\left(\frac{A}{\kappa_n B} + 1 - \exp(-\phi_{t,a})\left(\frac{A}{\kappa_n B} + 1 + \phi_{t,a} - \frac{\phi_{t,a}}{B}\right)\right)$ | (8) | $E_{t,a}$: annual transpiration (mm year⁻¹), $\phi_{t,a}$: an aridity index |
| $D_{i,d} = \min(S_{max}, E_{p,d})$ | (9) | $S_{max}$: the daily interception storage capacity (mm day⁻¹) $E_{p,d}$: the daily potential evaporation, $E_{p,a}$: annual potential evaporation (mm year⁻¹) |
| $S_{max} \approx C_{max} = 0.935 + 0.498LAI - 0.00575LAI^2$ | (10) | LAI: Leaf Area Index derived from remote sensing images |
| $\phi_{i,m} = \frac{D_{i,d}}{\beta}$ | (11) | β: scaling factor |
| $\beta = \frac{P_m}{E(n_{r,d}\|n_m)}$ | (12) | $E(n_{r,d}\|n_m)$: the expected number of rain days per month, $n_{r,d}$: the number of rain days per month, $n_m$: the number of days per month |
| $\phi_{i,a} = \frac{n_{r,d}D_{i,d}}{\kappa_m}$ | (13) | $\kappa_m$: scaling factor for monthly rainfall |
| $\kappa_m = \frac{P_a}{E(n_{r,m}\|n_a)}$ | (14) | $E(n_{r,m}\|n_a)$: the expected number of rain months per year, $n_{r,m}$: the number of rain months per year, $n_a$: the number of months per year |
| $B = 1 - \gamma + \gamma\exp\left(-\frac{1}{\gamma}\right)$ | (15) | $\gamma$: time scale for transpiration |
| $\gamma = \frac{S_b}{D_{t,m}\Delta t_m}$ | (16) | $S_b$: the moisture content below which transpiration is restricted (mm). |
| $D_{t,m} = 0 \quad for \ LAI < 0.1$ $D_{t,m} = \frac{E_p}{n_a}(-0.21 + 0.7LAI^{0.5}) \quad for \ 0.1 \le LAI < 2.7$ $D_{t,m} = \frac{E_p}{n_a} \quad for \ LAI \ge 2.7$ | (17) | $E_p$: annual potential evaporation (for open water) (mm year⁻¹) |
| $\phi_{t,a} = \frac{D_{t,m}}{\kappa_n}$ | (18) | $\kappa_n$: scaling factor for monthly net rainfall |
| $\kappa_n = \frac{P_{n,a}}{E(n_{nr,m}\|n_a)} = \frac{P_a - E_{i,a}}{E(n_{nr,m}\|n_a)}$ | (19) | $P_{n,a}$: annual net precipitation, $E(n_{nr,m}\|n_a)$: the expected number of net rain months per year |





**Table 3-** Comparison of mean annual evaporation estimated by Gerrits' model to Landflux-EVAL, STEAM and GLEAM through Average, RMSE, MBE and RE per land cover type. Negative MBE and RE show the Gerrits' model underestimates evaporation and vice versa. Average, RMSE and MBE are in mm year$^{-1}$ and RE is in %.

| Land cover | area | Gerrits | Landflux-EVAL | | | | STEAM | | | | GLEAM | | | |
|---|---|---|---|---|---|---|---|---|---|---|---|---|---|---|
| | 1000km² | Avg.* | Avg. | RMSE | MBE | RE | Avg. | RMSE | MBE | RE | Avg. | RMSE | MBE | RE |
| Evergreen needleleaf forest | 5563 | 430 | 387 | 122 | +43 | +10 | 467 | 150 | -37 | -9 | 457 | 127 | -27 | -6 |
| Evergreen broadleaf forest | 11778 | 1367 | 1177 | 266 | +190 | +14 | 1129 | 345 | +238 | +17 | 1244 | 225 | +123 | +9 |
| Deciduous needleleaf forest | 2498 | 338 | 298 | 73 | +40 | +12 | 336 | 65 | +2 | +1 | 336 | 73 | +1 | 0 |
| Deciduous broadleaf forest | 1106 | 796 | 736 | 138 | +61 | +8 | 840 | 215 | -44 | -6 | 660 | 197 | +136 | +17 |
| Mixed forest | 13470 | 563 | 487 | 136 | +76 | +13 | 545 | 137 | +18 | +3 | 527 | 131 | +35 | +6 |
| Shrublands[1] | 29542 | 203 | 259 | 96 | -57 | -28 | 262 | 123 | -59 | -29 | 253 | 91 | -51 | -25 |
| Savannas[2] | 18846 | 695 | 739 | 148 | -44 | -6 | 737 | 186 | -42 | -6 | 705 | 154 | -10 | -1 |
| Grasslands | 21844 | 275 | 365 | 130 | -91 | -33 | 373 | 164 | -98 | -36 | 349 | 135 | -75 | -27 |
| Croplands | 12417 | 488 | 535 | 124 | -47 | -10 | 583 | 209 | -95 | -20 | 486 | 118 | +2 | 0 |
| Croplands and natural vegetation mosaic | 5782 | 687 | 696 | 157 | -9 | -1 | 702 | 175 | -15 | -2 | 663 | 158 | +24 | +3 |
| Global[3] | - | 443 | 469 | - | - | -6 | 475 | - | - | -7 | 462 | - | - | -4 |

[1]including open and closed shrublands. [2]including woody savannas and savannas. [3]for overlapped pixels with 1.5°×1.5° resolution.





**Table 4-** Comparison of interception estimated by Gerrits' model to STEAM and GLEAM through Average, RMSE, MBE and RE per land cover type. Negative MBE and RE show the Gerrits' model underestimates evaporation and vice versa. Average, RMSE and MBE are in mm year[-1] and RE is in %.

| Land cover | Area | Gerrits | STEAM | | | | GLEAM | | | |
|---|---|---|---|---|---|---|---|---|---|---|
| | 1000km² | Avg. | Avg. | RMSE | MBE | RE | Avg. | RMSE | MBE | RE |
| Evergreen needleleaf forest | 5563 | 145 | 204 | 70 | -58 | -40 | 127 | 58 | +18 | +12 |
| Evergreen broadleaf forest | 11778 | 452 | 499 | 120 | -47 | -10 | 340 | 130 | +111 | +25 |
| Deciduous needleleaf forest | 2498 | 104 | 156 | 56 | -53 | -51 | 29 | 76 | +74 | +72 |
| Deciduous broadleaf forest | 1106 | 179 | 299 | 145 | -120 | -67 | 80 | 117 | +99 | +55 |
| Mixed forest | 13470 | 172 | 220 | 59 | -48 | -28 | 127 | 66 | +45 | +26 |
| Shrublands[1] | 29542 | 69 | 116 | 63 | -47 | -68 | 64 | 64 | +5 | +7 |
| Savannas[2] | 18846 | 162 | 246 | 107 | -84 | -52 | 107 | 79 | +55 | +34 |
| Grasslands | 21844 | 76 | 146 | 83 | -70 | -93 | 97 | 58 | -22 | -29 |
| Croplands | 12417 | 116 | 174 | 89 | -58 | -50 | 97 | 55 | +19 | +16 |
| Croplands and natural vegetation mosaic | 5782 | 166 | 243 | 108 | -77 | -46 | 112 | 89 | +54 | +33 |
| Global[3] | - | 128 | 183 | - | - | -44 | 109 | - | - | +15 |

[1]including open and closed shrublands. [2]including woody savannas and savannas. [3]for overlapped pixels with 1.5°×1.5° resolution.





**Table 5-** Comparison of transpiration estimated by Gerrits' model to STEAM and GLEAM through Average, RMSE, MBE and RE per land cover type. Negative MBE and RE show the Gerrits' model underestimates evaporation and vice versa. Average, RMSE and MBE are in mm year$^{-1}$ and RE is in %.

| Land cover | Area | Gerrits | STEAM | | | | GLEAM | | | |
|---|---|---|---|---|---|---|---|---|---|---|
| | 1000km² | Avg. | Avg. | RMSE | MBE | RE | Avg. | RMSE | MBE | RE |
| Evergreen needleleaf forest | 5563 | 284 | 222 | 122 | +63 | +22 | 259 | 100 | +25 | +9 |
| Evergreen broadleaf forest | 11778 | 915 | 619 | 347 | +296 | +32 | 890 | 163 | +25 | +3 |
| Deciduous needleleaf forest | 2498 | 234 | 177 | 82 | +57 | +24 | 261 | 71 | -21 | -12 |
| Deciduous broadleaf forest | 1106 | 617 | 538 | 192 | +79 | +13 | 570 | 120 | +47 | +16 |
| Mixed forest | 13470 | 390 | 305 | 147 | +85 | +22 | 363 | 114 | +27 | +7 |
| Shrublands[1] | 29542 | 133 | 137 | 85 | +4 | +3 | 159 | 81 | -26 | -20 |
| Savannas[2] | 18846 | 533 | 473 | 162 | +59 | +11 | 577 | 148 | -44 | -8 |
| Grasslands | 21844 | 199 | 214 | 109 | +15 | +7 | 233 | 93 | -34 | -17 |
| Croplands | 12417 | 372 | 393 | 131 | -20 | -5 | 371 | 90 | +1 | 0 |
| Croplands and natural vegetation mosaic | 5782 | 521 | 444 | 159 | +77 | +15 | 530 | 112 | -10 | -2 |
| Global[3] | - | 315 | 276 | - | - | +12 | 329 | - | - | -4 |

[1]including open and closed shrublands. [2]including woody savannas and savannas. [3]for overlapped pixels with 1.5°×1.5° resolution.


**Table 6–** Comparison of mean annual evaporation estimated by Gerrits' model to Schreiber, Ol'dekop, Pike and Budyko through Average, RMSE, MBE and RE per land cover type. Negative MBE and RE show the Gerrits' model underestimates evaporation and vice versa. Average, RMSE and MBE are in mm year$^{-1}$ and RE is in %.

| Land cover | area | Gerrits | Schreiber | | | | Ol'dekop | | | | Pike | | | | Budyko | | | |
|---|---|---|---|---|---|---|---|---|---|---|---|---|---|---|---|---|---|---|
| | 1000km² | Avg. | Avg. | RMSE | MBE | RE | Avg. | RMSE | MBE | RE | Avg. | RMSE | MBE | RE | Avg. | RMSE | MBE | RE |
| Evergreen needleleaf forest[1] | 5563 | 430 | 348 | 136 | +82 | +19 | 415 | 110 | +14 | +3 | 387 | 117 | +43 | +10 | 380 | 119 | +50 | +12 |
| Evergreen broadleaf forest | 11778 | 1367 | 876 | 526 | +491 | +36 | 1065 | 355 | +301 | +22 | 991 | 419 | +375 | +27 | 966 | 443 | +401 | +29 |
| Deciduous needleleaf forest | 2498 | 338 | 250 | 110 | +87 | +26 | 291 | 85 | +47 | +14 | 273 | 94 | +64 | +19 | 270 | 96 | +68 | +20 |
| Deciduous broadleaf forest | 1106 | 796 | 636 | 22 | +161 | +20 | 727 | 120 | +69 | +9 | 687 | 152 | +109 | +14 | 680 | 160 | +117 | +15 |
| Mixed forest | 13470 | 563 | 420 | 185 | +142 | +25 | 506 | 134 | +56 | +10 | 470 | 150 | +92 | +16 | 461 | 156 | +101 | +18 |
| Shrublands[1] | 29542 | 203 | 250 | 84 | -48 | -24 | 273 | 99 | -71 | -35 | 263 | 91 | -60 | -30 | 261 | 90 | -59 | -29 |
| Savannas[2] | 18846 | 695 | 648 | 168 | +47 | +7 | 757 | 167 | -62 | -9 | 710 | 155 | -15 | -2 | 700 | 155 | -5 | -1 |
| Grasslands | 21844 | 275 | 346 | 134 | -71 | -26 | 372 | 152 | -98 | -36 | 359 | 141 | -85 | -31 | 358 | 140 | -84 | -31 |
| Croplands | 12417 | 488 | 502 | 154 | -14 | -3 | 566 | 181 | -78 | -16 | 538 | 164 | -50 | -10 | 533 | 162 | -45 | -9 |
| Croplands and natural vegetation mosaic | 5782 | 687 | 617 | 221 | +69 | +10 | 721 | 195 | -35 | -5 | 677 | 196 | -10 | -1 | 667 | 200 | -20 | -3 |
| Global[3] | - | 443 | 410 | - | - | +8 | 471 | - | - | -6 | 445 | - | - | 0 | 439 | - | - | +1 |

[1]including open and closed shrublands. [2]including woody savannas and savannas. [3]for overlapped pixels with 1.5°×1.5° resolution.





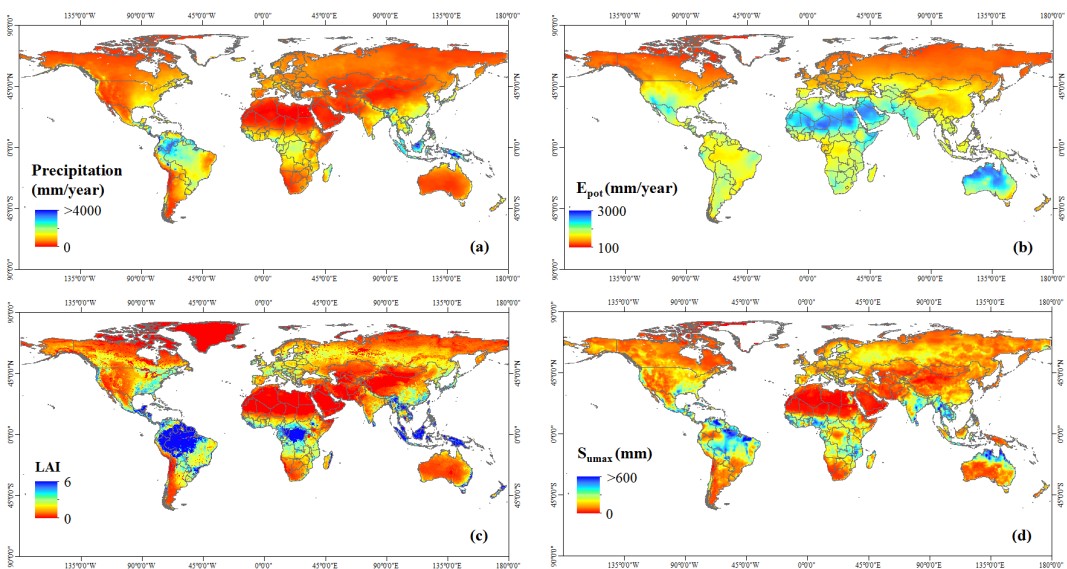

**Figure 1-** Mean annual of the applied data in the current study: (a) Precipitation (Ruane et al.,
2015), (b) Potential evaporation (University of East Anglia Climatic Research Unit, 2014), (c) LAI
(Zhu et al., 2013) and (d) $S_{u,max}$ (Wang-erlandsson et al., 2016).





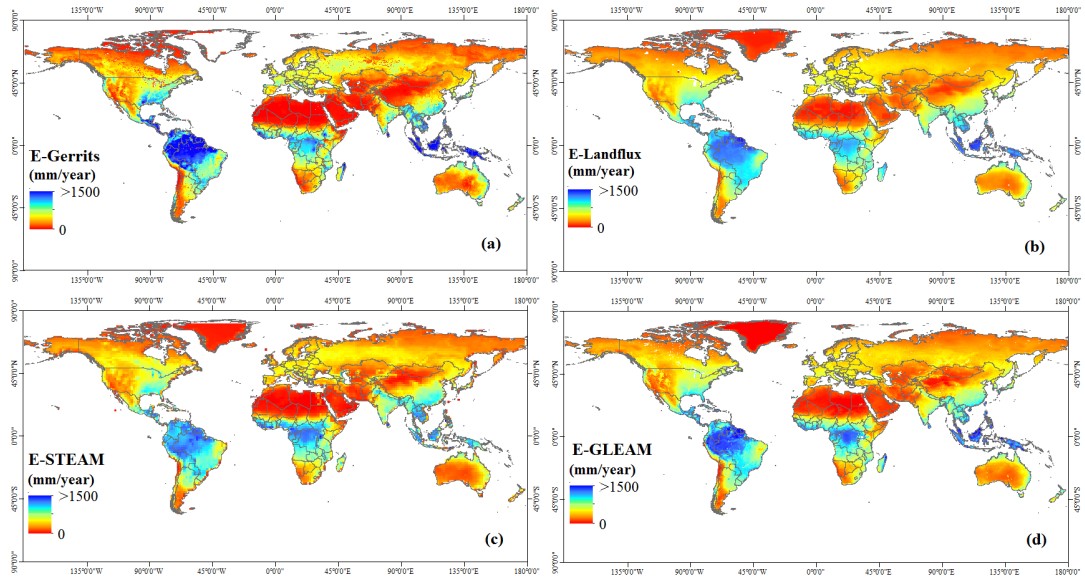

**Figure 2-** Mean annual evaporation estimated by (a) Gerrits' model, (b) Landflux-EVAL, (c) STEAM and (d) GLEAM.





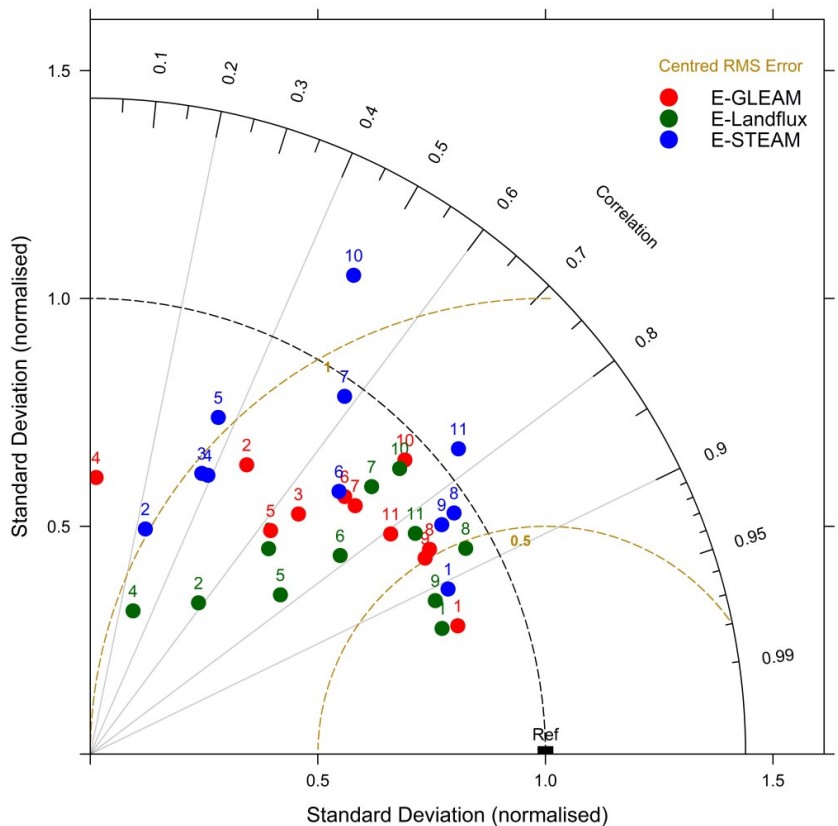

**Figure 3-** Taylor diagram for mean annual evaporation estimated by Gerrits' model in comparison
to Landflux-EVAL (green circles), STEAM (blue circles) and GLEAM (red circles) for all data
(No. 1), Evergreen Needleleaf Forest (No.2), Evergreen broadleaf forest (No. 3), Deciduous
needleleaf forest (No. 4), Deciduous broadleaf forest (No. 5), Mixed Forest (No. 6), Shrublands
(No. 7), Savannas (No. 8), Grasslands (No. 9), Croplands (No. 10) and Croplands and natural
vegetation mosaic (No. 11).





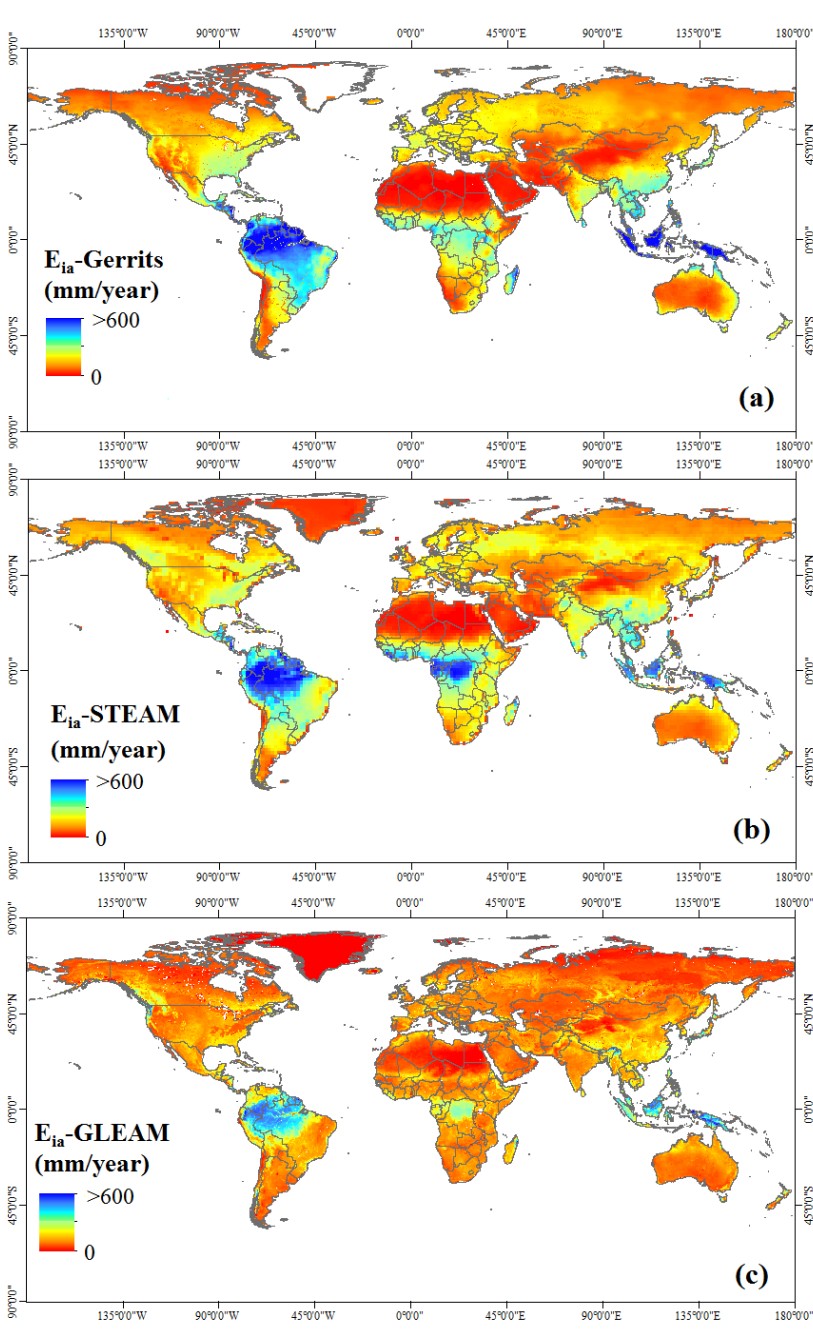

**Figure 4-** Simulated mean annual interception by (a) Gerrits' model and (b) STEAM and (c) GLEAM.



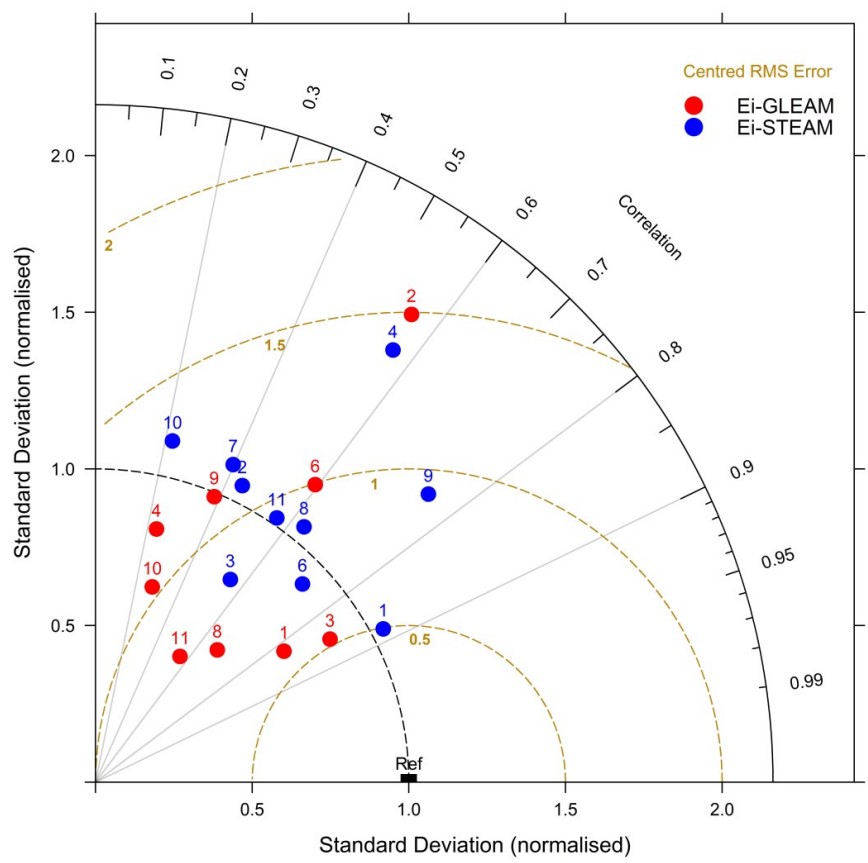

**Figure 5-** Taylor diagram for mean annual interception estimated by Gerrits' model in comparison
to STEAM (blue circles) and GLEAM (red circles) for all data (No. 1), Evergreen Needleleaf
Forest (No.2), Evergreen broadleaf forest (No. 3), Deciduous needleleaf forest (No. 4), Deciduous
broadleaf forest (No. 5), Mixed Forest (No. 6), Shrublands (No. 7), Savannas (No. 8), Grasslands
(No. 9), Croplands (No. 10) and Croplands and natural vegetation mosaic (No. 11).





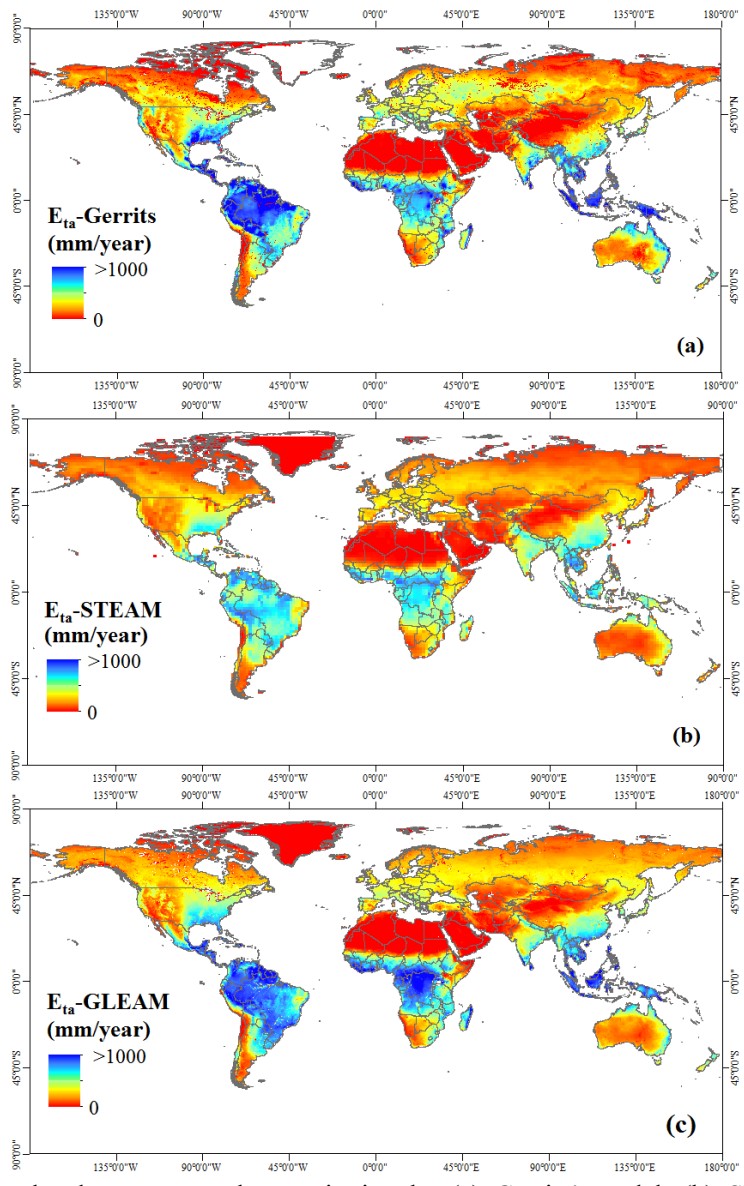

2  **Figure 6-** Simulated mean annual transpiration by (a) Gerrits' model, (b) STEAM and (c)
3  GLEAM.





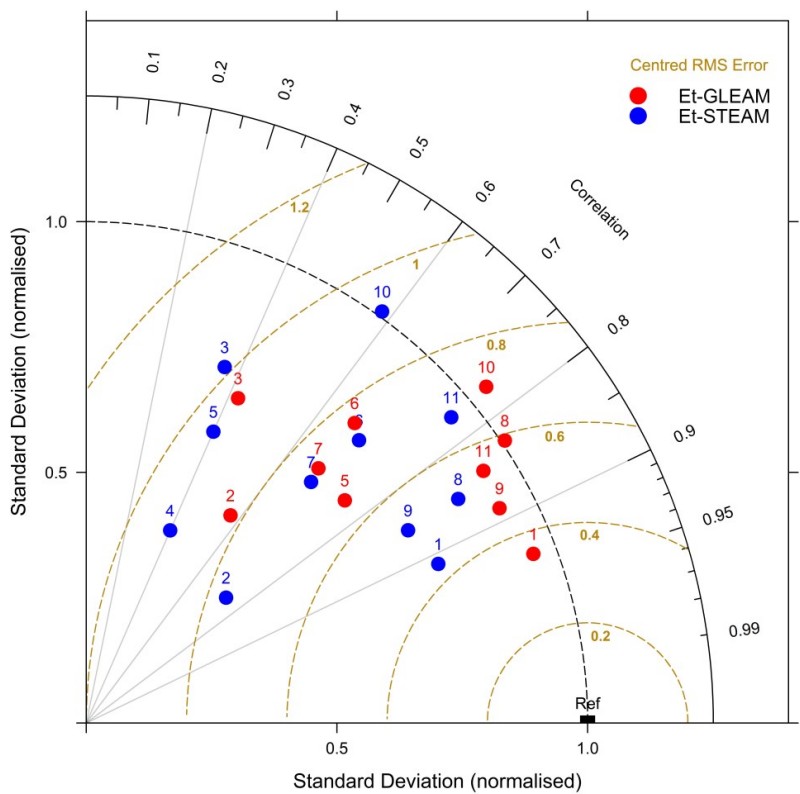

**Figure 7-** Taylor diagram for mean annual transpiration estimated by Gerrits' model in comparison
to STEAM (blue circles) and GLEAM (red circles) for all data (No. 1), Evergreen Needleleaf
Forest (No.2), Evergreen broadleaf forest (No. 3), Deciduous needleleaf forest (No. 4), Deciduous
broadleaf forest (No. 5), Mixed Forest (No. 6), Shrublands (No. 7), Savannas (No. 8), Grasslands
(No. 9), Croplands (No. 10) and Croplands and natural vegetation mosaic (No. 11).

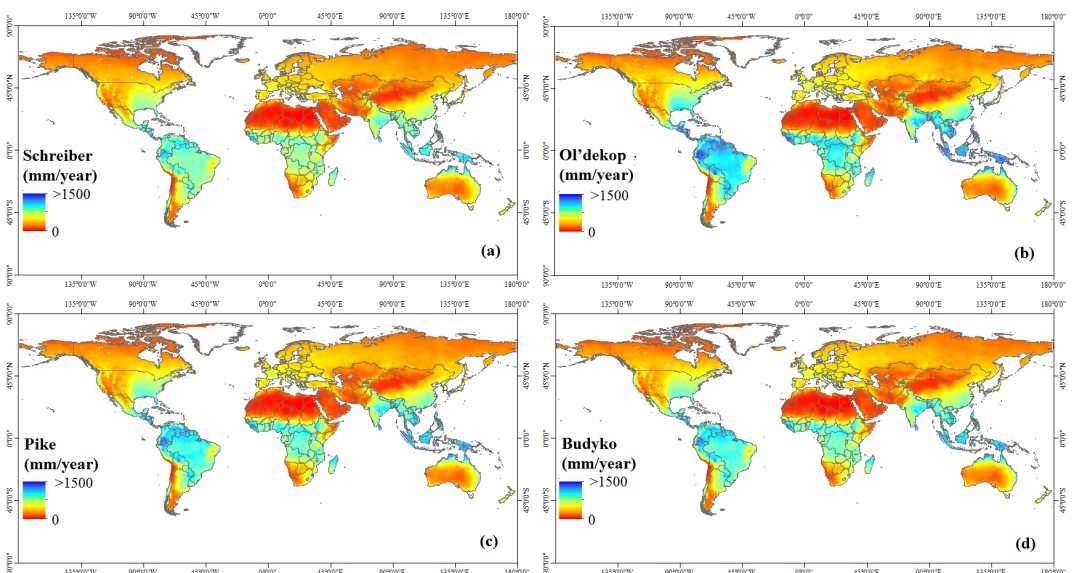

2    **Figure 8-** Global evaporation (mm year[-1]) estimated by Budyko curves: (a) Schreiber (1904), (b)
3    Ol'dekop (1911), (c) Pike (1964), and (d) Budyko (1974).





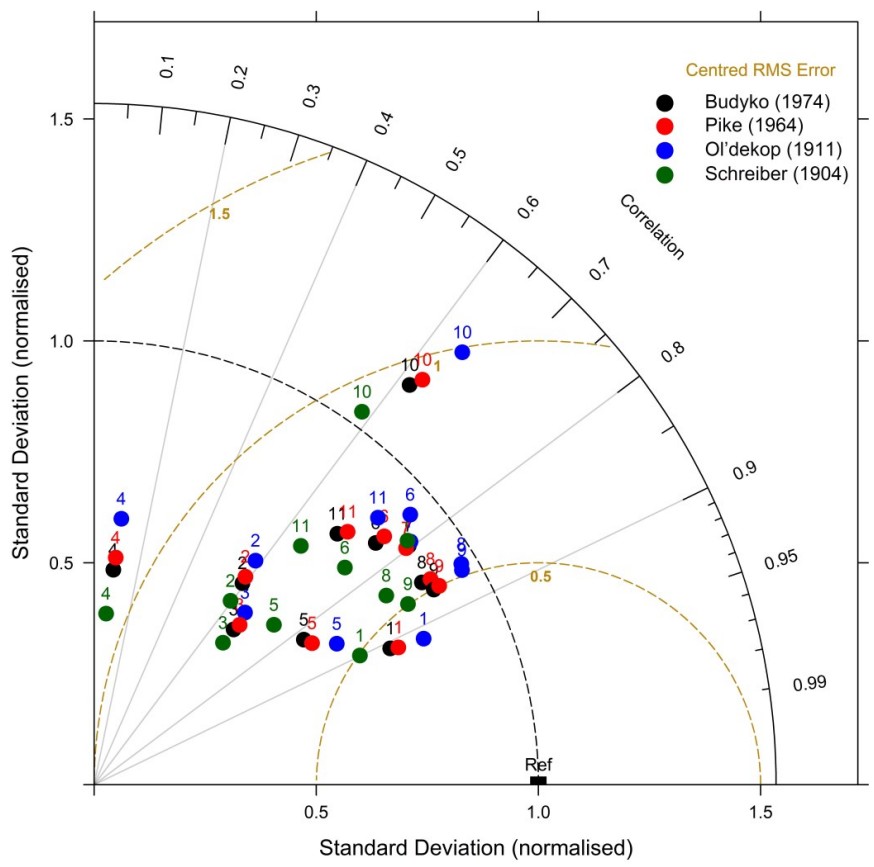

**Figure 9-** Taylor diagram for mean annual evaporation estimated by Gerrits' model in
comparison to Schreiber (1904) (green circles), Ol'dekop (1911) (blue circles), Pike (1964) (red
circles), and Budyko (1974) (black circles) for all data (No. 1), Evergreen Needleleaf Forest
(No.2), Evergreen broadleaf forest (No. 3), Deciduous needleleaf forest (No. 4), Deciduous
broadleaf forest (No. 5), Mixed Forest (No. 6), Shrublands (No. 7), Savannas (No. 8), Grasslands
(No. 9), Croplands (No. 10) and Croplands and natural vegetation mosaic (No. 11).



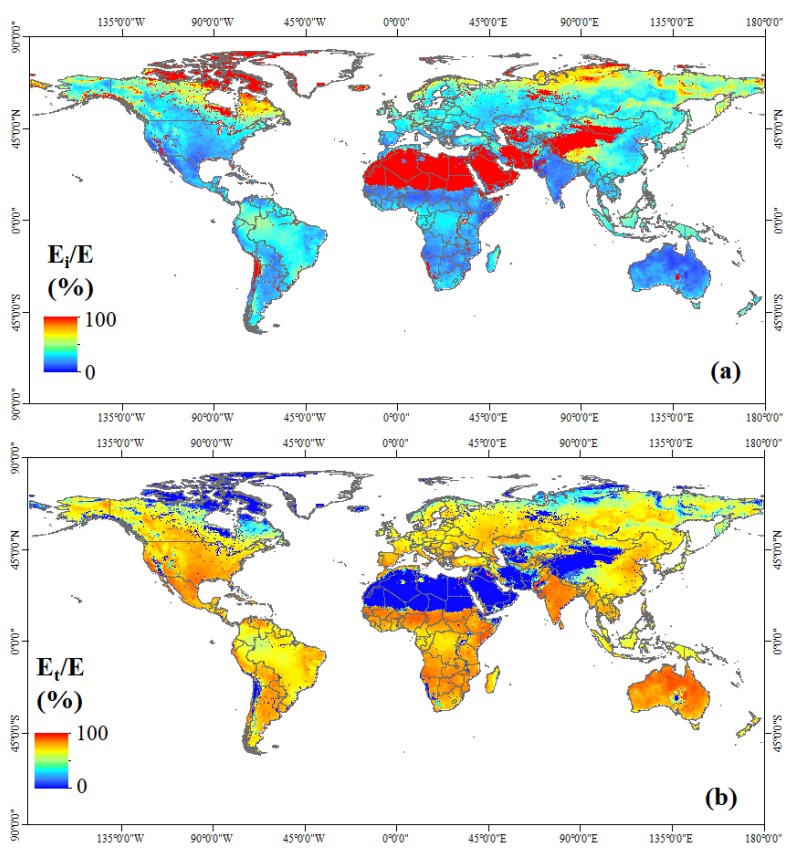

2    **Figure 10-** (a) Interception and (b) Transpiration ratio as a percentage of mean annual evaporation.