# Peer review of "A global Budyko model to partition evaporation into interception and transpiration"

_Hydrology and Earth System Sciences, 2018_

## Referee Comment (RC1) · Stephen Good (Referee) · 24 Apr 2019

GENERAL ASSESSMENT:

As shown by the recent multi-model assessment by Miralles (HESS 2016) or by Kumar (Remote Sensing 2018), contemporary estimates of the evapotranspiration subcomponents from models are wildly different. In many of these inter-comparisons, total evapotranspiration flux is relatively consistent between models while the divergence in the sub-components is large. This submitted paper presents another model, based on the Gerrits (WRR 2009) approach, that provides partitioned estimates of surface to atmosphere water flux. Results from this new model are compared with a few other models (GLEAM & STEAM), with differences between models broken down by land cover

type. This modeling and analysis are conducted in a satisfactory manner. However, it is hard to see how yet one more model that estimates evapotranspiration subcomponents moves us closer to a better understanding of these fluxes.

The introduction and a paragraph in the discussion relate this model to the Budyko framework. One possible way forward for the authors is evaluating how trends in flux components relate the energy and water limitations outlined by the Budyko framework, since this is the stated motivation of this model. This could move the paper beyond how it is currently presented as another land surface model applied using remote sensing observations. For example, see Figure 11 of Miralles's 2016 HESS paper for casting total evaporative fluxes in this context. Also relevant is the study of Good (Nature Ecology & Evolution 2018) which used a Budyko approach to examine how to partition evaporative fluxes. In revising the paper, I suggest the authors work to find how this approach helps us understand the different surface to atmosphere water flux pathways better.

Most critically, I find the language in this paper to be grandiose and predicated on a poorly based argument. As is written in the abstract and introduction, the authors suggest that others have "tried to improve the Budyko framework by including more physics and catchment characteristics . . . However this often resulted in additional parameters, which are unknown or difficult to determine." This statement, and others like it in this paper, is inappropriate for two reasons: (1) other approaches have used fairly easy to measure characteristics and (2) because the authors proceed to do exactly what they claim shouldn't be done by fitting "difficult" to determine parameters to optimize their results. For point (1) for instance, the approach of Porporato is explicitly physically based as is it dependent on the ratio soil water storage to mean rainfall depth which is a measurable quantity. Furthermore, both of these quantities are used in the analysis presented here. For reason (2), the 'b' parameter of this analysis, among others, is clearly stated by the authors (P5L15) to have been calibrated to produce the best results. This is very similar to the Li (WRR 2013) paper wherein the Budyko

curve parameters were fit to vegetation cover. The authors use of language such as "tried" (P2L18) seems to imply these other authors were unsuccessful, which may not be true. In my opinion, this submitted paper is quite similar to these other efforts in that it has extended the Budyko framework with new parameters they have fitted based on physical processes. Here, the most important parameters dictating the transpiration component are when transpiration becomes downregulated, and how much maximal transpiration can be. Equation 17 needs more elaboration and justification, as does the parameterization of Sb as 50% of $S_{u,max}$. How were these values selected and what is the consequence of other using other values here. How much do these choices, and other values such as the 'b' parameter, influence model outcomes.

SPECIFIC COMMENTS:

P1L11: The $1/(1+f(phi))$ is not the base of all Budyko curves. Budyko, himself used a hyperbolic tangent as an example. What do the lower and upper case f's represent? P2L33: This paper estimates available soil water capacity, not the actual soil water itself. Also, I wouldn't call these 'data' but modeled estimates. P3L16: Evaporation from 'non-superfical' soil moisture P4L11: Do you mean daily, not yearly, average? P5L14: I think you should also place these eqn in table 2 for consistency: $A = b*S_{u,max}$ as well as $Sb = 0.5*S_{u,max}$ P5L36: Reword here. As is stated above and in eq17, you do not hold $D_{t,m}$ constant? Which is it? P5L38: Do you have a justification or citation for this statement? P7L17: No observations where used here. Only comparisons of the Gerrits model against other models. P8L42: There are many bare soil estimates (See the review by Kool 2014 Agg and Forest Met, for example). F2: Because of the size of these figures, and the large range of values, it becomes hard to discern differences. Why not plot the absolute value of E flux in panel A, and then the differences in panels B, C, and D. Consider this approach in later figures as well F3: Units for the RMSE, here and onward.

---

## Referee Comment (RC2) · Anonymous Referee #2 · 29 Apr 2019

The Gerrits' model is applied to on the global scale using remote sensing data estimated maximum root zone storage capacity. The results are compared with two land surface model outputs. The results of the paper provide the global distributions of interception and transpiration. I have a few comments for the authors to consider for revision. Equation (3b): $E = E_i + E_t$. Since $E_i$ includes soil evaporation, I would suggest to interpret this as $ET = E + T$ where $E$ is evaporation and $T$ is transpiration. Page 3 Line 21: Does $E_t$ have the same definition as the $E_s$ defined in Line 16? Page 4 Lines 28-30: whys accounting for n is rarely necessary? Maybe it is better to explain it briefly here. Page 5 Line 36: If the inter-annual variability of the $D_{t,m}$ has any impact on the results? Page 5 Lines 37-37: "But in those relatively wet areas transpiration is underestimated in summer, but overestimated in winter, which will cancel out on the

annual scale." Delete the first "But"? Page 7 Line 32: year-1 Page 8 Lines 2-3: Is there any analysis in this study to demonstrate that the precipitation is the major factor that caused the different results from different models? Page 9 Lines 27-32: The global transpiration ratio estimated by Gerrits' model is larger than nearly all of the other studies listed, is there any reason? Page 10 Lines 27-29: Since the constant value of 0.935 mm in Equation 10 could be underestimated for the forest floor interception, then what value is advised for the forest floor?

---

## Author Comment (AC1) · 27 May 2019

Dear Stephen Good,

We really appreciate your detailed review on our manuscript. The reactions to the comments are as follows.

**General Assessment**

**Comment 1:**

**This modeling and analysis are conducted in a satisfactory manner. However, it is hard to see how yet one more model that estimates evapotranspiration subcomponents moves us closer to a better understanding of these fluxes.**

**Reaction:**

Our aim is not to provide yet another LSM that partitions evaporation. Our aim is to show with a simple analytical model that the Budyko framework can be explained. For this we use the reasoning of the Gerrits model that recognizes the characteristic time scales of the different evaporation processes (i.e. interception daily and transpiration monthly). We revised the Gerrits-model in such way that it was possible to apply it at the global scale. As suggested by the reviewer, we will clarify this better and will relate the results of the model to the Budyko framework for a better understanding the partitioning of evaporation into transpiration and interception.

**Comment 2:**

**The introduction and a paragraph in the discussion relate this model to the Budyko framework. One possible way forward for the authors is evaluating how trends in flux components relate the energy and water limitations outlined by the Budyko framework, since this is the stated motivation of this model. This could move the paper beyond how it is currently presented as another land surface model applied using remote sensing observations. For example, see Figure 11 of Miralles's 2016 HESS paper for casting total evaporative fluxes in this context. Also relevant is the study of Good (Nature Ecology & Evolution 2018) which used a Budyko approach to examine how to partition evaporative fluxes. In revising the paper, I suggest the authors work to find how this approach helps us understand the different surface to atmosphere water flux pathways better.**

**Reaction:**

We thank the reviewer for this valuable suggestion. We agree that our aim was not clearly defined and also misleading in that sense. As suggested by the reviewer, we will evaluate the relation between evaporation fluxes and energy/water limitation in Budyko framework as

provided by Miralles et al. (2016) and Good et al. (2017). In the graphs, we provide the Budyko framework for each land cover, and for the evaporation fluxes ($E_i$ and $E_t$ and $E_{tot}$), separately, to discuss how our model can be related to Budyko framework and how the energy and water limitation can be interpreted by our model.

**Comment 3:**

**Most critically, I find the language in this paper to be grandiose and predicated on a poorly based argument. As is written in the abstract and introduction, the authors suggest that others have "tried to improve the Budyko framework by including more physics and catchment characteristics… However this often resulted in additional parameters, which are unknown or difficult to determine." This statement, and others like it in this paper, is inappropriate for two reasons: (1) other approaches have used fairly easy to measure characteristics and (2) because the authors proceed to do exactly what they claim shouldn't be done by fitting "difficult" to determine parameters to optimize their results. For point (1) for instance, the approach of Porporato is explicitly physically based as is it dependent on the ratio soil water storage to mean rainfall depth which is a measurable quantity. Furthermore, both of these quantities are used in the analysis presented here. For reason (2), the 'b' parameter of this analysis, among others, is clearly stated by the authors (P5L15) to have been calibrated to produce the best results. This is very similar to the Li (WRR 2013) paper wherein the Budyko curve parameters were fit to vegetation cover. The authors use of language such as "tried" (P2L18) seems to imply these other authors were unsuccessful, which may not be true. In my opinion, this submitted paper is quite similar to these other efforts in that it has extended the Budyko framework with new parameters they have fitted based on physical processes. Here, the most important parameters dictating the transpiration component are when transpiration becomes downregulated, and how much maximal transpiration can be. Equation 17 needs more elaboration and justification, as does the parameterization of Sb as 50% of S_u,max. How were these values selected and what is the consequence of other using other values here. How much do these choices, and other values such as the 'b' parameter, influence model outcomes.**

**Reaction:**

Yes, you are right that we also have some calibration parameters and should therefore rephrase our text. Nonetheless, we think that we use a slightly different approach for these calibration parameters and other model parameters as well. Although others indeed also use 'measurable parameters', which could be tested in some case studies, some of these input values are not available at the global scale as for example the soil water storage. For example, carry over parameter ($A$) was available for 10 locations in Gerrits et al. (2009), but at global scale we did not have such data, so we proposed $A=b*S_{u,max}$, and we need to calibrate the "$b$" parameter to link $A$ to a measurable variable. So, yes, we admit that you are right and we need to rephrase this part of the manuscript.

About the $S_b$ as 50% of $S_{u,max}$, we mentioned in the text that in this study we assumed $S_b$ to be 50% of $S_{u,max}$, as this value is commonly used for many crops, referred to (Allen et al. 1998). We will provide a sensitivity analysis in the revised manuscript to see how our results would change in response to the changes in the parameters.

**Specific Comments:**

**Comment 4:**

**P1L11: The 1/(1+f(phi)) is not the base of all Budyko curves. Budyko, himself used a hyperbolic tangent as an example. What do the lower and upper case f's represent?**

**Reaction:**

As mentioned by Arora (2002), evaporation ratio ($E/P$) is a function of the aridity index ($\Phi$) and Bowen ratio ($\gamma$) ($\frac{E}{P} = \frac{\emptyset}{1+\gamma}$). Arora interpreted the equation as follows:

"As a region becomes dry and is characterized by high potential evaporation, low precipitation and evapotranspiration, and high sensible heat fluxes then $\Phi \rightarrow 1$, $\gamma \rightarrow 1$ and $E/P$ tends towards unity implying little runoff. the other hand, as a region becomes wet and is characterized by low values of aridity index ($\Phi$) and Bowen ratio ($\gamma$) then $E/P < 1$ and runoff occurs. Since Bowen ratio ($\gamma$) is also a function of available energy and precipitation (and thus a function of $\Phi$) evaporation ratio may be expressed as a function of aridity index alone."

It leads to equation 1 in our paper. Thus, in equation 1, f and F are both mathematical functions, showing that $E/P$ is a function of the aridity ($\Phi$). F($\Phi$) can have many forms (exponential, hyperbolic tangent, etc.) as summarized in Table 1.

**Comment 5:**

**P2L33: This paper estimates available soil water capacity, not the actual soil water itself. Also, I wouldn't call these 'data' but modeled estimates.**

**Reaction:**

Gao et al. (2014) presented a new method where the available soil water is derived from time series of rainfall and potential evaporation, plus a long-term runoff coefficient. We agree that knowing soil moisture storage change is important for Budyko framework, but we use a method whereby we work around it by using plant available water. The method of Gao et al. (2014) provides plant available water (which is often linked to soil water capacity). In our paper we

used it as $S_{u,max}$. We will rephrase it in the manuscript to explain it more preciously. Moreover, we agree that it's better to call this 'modeled estimates' rather than 'data'.

**Comment 6:**

**P3L16: Evaporation from 'non-superficial' soil moisture**

**Reaction:**

Thanks. We will add this.

**Comment 7:**

**P4L11: Do you mean daily, not yearly, average.**

**Reaction:**

Yes, daily average during the year. We will correct it in the text.

**Comment 8:**

**P5L14: I think you should also place these eqn in table 2 for consistency: A = b\*S_u,max as well as Sb = 0.5\*S_u,max**

**Reaction:**

Ok, will be done.

**Comment 9:**

**P5L36: Reword here. As is stated above and in eq17, you do not hold Dt,m constant? Which is it?**

**Reaction:**

We keep $D_{t,m}$ constant during the year (like $D_{i,d}$), but equation 17 shows that we calculated it as a function of the average yearly LAI. For water-constrained areas this is not a problem, because there $E_{t,m}$ is determined by the LHS of the min-function ($A + B(P_m - E_{i,m})$) as can be seen in Equation 7. For energy-constrained areas our assumption can be problematic. However, in those areas often temperature and radiation follow a sinusoidal pattern without complex double

seasonality as e.g., occurs in the ITCZ. This implies that the overestimation of $E_{t,m}$ in winter will be compensated (on the annual time scale) by the underestimation in summer time.

**Comment 10:**

**P5L38: Do you have a justification or citation for this statement?**

**Reaction:**

Please see our response to comment 9.

**Comment 11:**

**P7L17: No observations where used here. Only comparisons of the Gerrits model against other models.**

**Reaction:**

It is a general sentence for Taylor diagram, not only for our model. But to make it clear, we change "This diagram can provide a concise..." into 'A Taylor diagram can provide a concise..."

**Comment 12:**

**P8L42: There are many bare soil estimates (See the review by Kool 2014 Agg and Forest Met, for example).**

**Reaction:**

We meant that there is hardly any data on the forest floor interception storage capacity ($S_{max}$). We did not intend to refer to bare soil evaporation.

**Comment 13:**

**F2: Because of the size of these figures, and the large range of values, it becomes hard to discern differences. Why not plot the absolute value of E flux in panel A, and then the differences in panels B, C, and D. Consider this approach in later figures as well**

**Reaction:**

Thanks for your suggestion. We did it before, but the single pixel outliers may blow up the entire figure what was also not a good way of showing the differences. That's why we moved towards the Taylor diagrams. Moreover, we also wanted to show the original data.

**Comment 14:**

**F3: Units for the RMSE, here and onward.**

**Reaction:**

We used normalized RMSE in these figures as shown in the following equation, so it has no unit.

$$\text{NRMSE} = \frac{\text{RMSE}}{\overline{X_O}} \tag{1}$$

In this equation, NRMSE is normalized RMSE and $\overline{X_O}$ is the average of observation values (here the values estimated by Gerrits' model).

**References**

Allen RG, Pereira LS, Raes D, Smith M (1998) Crop evapotranspiration-Guidelines for computing crop water requirements-FAO Irrigation and drainage paper 56. FAO56, Rome

Arora VK (2002) The use of the aridity index to assess climate change effect on annual runoff. J Hydrol 265:164–177. doi: 10.1016/S0022-1694(02)00101-4

Gao H, Hrachowitz M, Schymanski SJ, et al (2014) Climate controls how ecosystems size the root zone storage capacity at catchment scale. Geophys Res Lett 41:7916–7923. doi: 10.1002/2014GL061668

Gerrits AMJ, Savenije HHG, Veling EJM, Pfister L (2009) Analytical derivation of the Budyko curve based on rainfall characteristics and a simple evaporation model. Water Resour Res 45:W04403. doi: 10.1029/2008WR007308

Good SP, Moore GW, Miralles DG (2017) A mesic maximum in biological water use demarcates biome sensitivity to aridity shifts. Nat Ecol Evol 1:1883–1888. doi: 10.1038/s41559-017-0371-8

Miralles DG, Jiménez C, Jung M, et al (2016) The WACMOS-ET project - Part 2: Evaluation of global terrestrial evaporation data sets. Hydrol Earth Syst Sci 20:823–842. doi: 10.5194/hess-20-823-2016

---

## Author Comment (AC2) · 27 May 2019

The authors would like to express their sincere gratitude to the Anonymous Referee (Reviewer 2) for his/her useful comments. The reactions to the comments are as follows.

**Comment 1:**

**Equation (3b): E = Ei + Et. Since Ei includes soil evaporation, I would suggest to interpret this as ET = E +T where E is evaporation and T is transpiration.**

**Reaction:**

As suggested by Savenije (2004) and based on the definition of total evaporation provided by Shuttleworth (1993), we call the sum of interception ($E_i$), soil evaporation ($E_s$), transpiration ($E_t$), and evaporation from water bodies ($E_o$) as "evaporation" ($E$). Thus, we did not apply the term of "evapotranspiration" (ET), because we agree that they "are very different in terms of time scale, time of occurrence, physical characteristics, climatic feedback and isotope composition" as explained in Savenije (2004).

**Comment 2:**

**Page 3 Line 21: Does Et have the same definition as the Es defined in Line 16?**

**Reaction:**

Yes, as mentioned in reaction to comment 1, $E_t$ is transpiration which is evaporation from the soil moisture connected to the root zone, because that's where the trees get their water from.

**Comment 3:**

**Page 4 Lines 28-30: whys accounting for n is rarely necessary? Maybe it is better to explain it briefly here.**

**Reaction:**

Miralles et al. (2010) and Pearce and Rowe (1981) both mentioned that accounting for $n$ is rarely necessary. Pearce and Rowe (1981) mentioned that "In many climates, however, such adjustments will not be necessary, or small enough that they can be neglected". In our interpretation this is because the number or times the interception storage can be filled and completely emptied is limited once we assume a drying time of ca. 4 hours, which is common.

For 12 hours of day light, it means that $n$ can be maximal 3 times. However, the chance that you have 4 storms every 4 hours, with a drying period of 4 hours, is rather small for most climates.

**Comment 4:**

**Page 5 Line 36: If the inter-annual variability of the Dt,m has any impact on the results?**

**Reaction**

We explained in the manuscript (page 5, line 36-38) that taking a constant value for $D_{t,m}$ can be problematic in energy-constrained areas. For water-constrained areas this is not a problem, because there $E_{t,m}$ is determined by the LHS of the min-function ($A + B(P_m - E_{i,m})$) as can be seen in Equation 7. For energy-constrained areas our assumption can be problematic. However, in those areas often temperature and radiation follow a sinusoidal pattern without complex double seasonality as e.g., occurs in the ITCZ. This implies that the overestimation of $E_{t,m}$ in winter will be compensated (on the annual time scale) by the underestimation in summer time.

In addition, Gerrits et al. (2009) provided a sensitivity analysis on the effect of different $D_{t,m}$ on total evaporation. Their results showed that total evaporation is sensitive to $D_{t,m}$ only once the annual rainfall exceed ±1000 mm/y.

**Comment 5:**

**Page 5 Lines 37-37: "But in those relatively wet areas transpiration is underestimated in summer, but overestimated in winter, which will cancel out on the annual scale." Delete the first "But"?**

**Reaction**

Thanks, it will be done in the revised manuscript.

**Comment 6:**

**Page 7 Line 32: year-1**

**Reaction**

Thanks, it will be corrected in the revised manuscript.

**Comment 7:**

**Page 8 Lines 2-3: Is there any analysis in this study to demonstrate that the precipitation is the major factor that caused the different results from different models?**

**Reaction**

By providing a sensitivity analysis in the revised manuscript, we can show how the model is sensitive to precipitation. Moreover, the results of sensitivity analysis conducted by Gerrits et al. (2009) shows that the results are significantly sensitive to change in $n_{r,d}$. We revised the manuscript as follows:

"Different precipitation products applied in the models is likely the reason for the differences. As found by Gerrits et al. (2009), the sensitivity of the model to the number of rain days and rain months especially for the higher rates of precipitation can be a probable reason for poor performance of a model especially for evergreen forests with the highest amount of precipitation."

**Comment 8:**

**Page 9 Lines 27-32: The global transpiration ratio estimated by Gerrits' model is larger than nearly all of the other studies listed, is there any reason?**

**Reaction**

Our transpiration ratio estimate is indeed in the higher range compared to other models/studies, however, the transpiration ratio estimated by Miralles et al. (2011) is higher than our model (80% in comparison to 71%). Moreover, our estimation is close to that of Sutanto (2015) (69%) and Good et al. (2015) (65%). Coenders-Gerrits et al. (2014) also found that based on the model of Jasechko et al. (2013) transpiration ratio changes between 35% and 80%, which is in line with our current findings.

**Comment 9:**

**Page 10 Lines 27-29: Since the constant value of 0.935 mm in Equation 10 could be underestimated for the forest floor interception, then what value is advised for the forest floor?**

**Reaction**

Forest floor evaporation can be modeled for each region based on its characteristics (e.g., Wang-Erlandsson et al. (2014)). Or typical values on $S_{max}$ for the forest floor can be found in Table 22.1 of Gerrits and Savenije (2011). For example, in the UK for Pine (Pinus sylvestris), it is 0.6-1.7 mm (Walsh and Voigt, 1977), in Australia for Eucalyptus, it is 1.7 mm (Putuhena and Cordery, 1996) and in Luxembourg for Beech (Fagus sylvatica), it is 1-2.8 mm (Gerrits et al., 2010).

**References**

Coenders-Gerrits A M J, van der Ent R J, Bogaard T A, Wang-Erlandson L, Hrachowitz M and Savenije H H G (2014) Uncertainties in transpiration estimates. Nature *506*: E1–E2 Retrieved from http://dx.doi.org/10.1038/nature12925

Gerrits A M J, Pfister L and Savenije H H G (2010) Spatial and temporal variability of canopy and forest floor interception in a beech forest. Hydrological Processes *24*(21): 3011–3025

Gerrits A M J and Savenije H H G (2011) Forest Floor Interception. In D. F. Levia, D. E. Carlyle-Moses & T. Tanaka (eds.), Forest Hydrology and Biogeochemistry: Synthesis of Past Research and Future Directions (Vol. 216). Ecological Studies Series, No. 216, Springer-Verlag,: Heidelberg, Germany Retrieved from http://link.springer.com/10.1007/978-94-007-1363-5

Gerrits A M J, Savenije H H G, Veling E J M and Pfister L (2009) Analytical derivation of the Budyko curve based on rainfall characteristics and a simple evaporation model. Water Resources Research *45*: W04403 Retrieved from http://doi.wiley.com/10.1029/2008WR007308

Good S P, Noone D and Bowen G (2015) Hydrologic connectivity constrains partitioning of global terrestrial water fluxes. Science *349*(6244): 175–177

Jasechko S, Sharp Z D, Gibson J J, Birks S J, Yi Y and Fawcett P J (2013) Terrestrial water fluxes dominated by transpiration. Nature *496*(7445): 347–350

Miralles D G, De Jeu R A M, Gash J H, Holmes T R H and Dolman A J (2011) Magnitude and variability of land evaporation and its components at the global scale. Hydrology and Earth System Sciences *15*: 967–981

Miralles D G, Gash J H, Holmes T R H, De Jeu R A M and Dolman A J (2010) Global canopy interception from satellite observations. Journal of Geophysical Research Atmospheres *115*(16): 1–8

Pearce A J and Rowe L K (1981) Rainfall interception in a multi-storied, evergreen mixed forest: estimates using Gash's analytical model. Journal of Hydrology *49*: 341–353

Putuhena W M and Cordery I (1996) Estimation of interception capacity of the forest floor. Journal of Hydrology *180*(1–4): 283–299 Retrieved from https://linkinghub.elsevier.com/retrieve/pii/0022169495028838

Savenije H H G (2004) The importance of interception and why we should delete the term evapotranspiration from our vocabulary. Hydrological Processes *18*(8): 1507–1511

Shuttleworth W J (1993) Evaporation. In Handbook of Hydrology. McGraw-Hill, New York,

Sutanto S J (2015) Global transpiration fraction derived from water isotopologue datasets. Jurnal Teknik Hidraulik *6*(2): 131–146

Walsh R P D and Voigt P J (1977) Vegetation Litter: An Underestimated Variable in Hydrology

and Geomorphology. Journal of Biogeography *4*(3): 253 Retrieved from https://www.jstor.org/stable/3038060?origin=crossref

Wang-Erlandsson L, Van Der Ent R J, Gordon L J and Savenije H H G (2014) Contrasting roles of interception and transpiration in the hydrological cycle - Part 1: Temporal characteristics over land. Earth System Dynamics *5*(2): 441–469

---

## Author Response (AR1)

Dear Editor,

We provided the replies to the comments as follows, with referring to the probable changes in the revised manuscript. In addition to the changes from reviewers' comments, we removed Figures 1 and 10. Moreover, we removed Figures 8 and 9 and table 6 (comparing to the Budyko curves), because we evaluated our results through the Budyko framework, as suggested by Stephen Good. Accordingly, any explanations in the text related to these figures and table were also removed. These changes are shown in the manuscript by track changes.

**Reviewer 1 (Stephen Good):**

**General Assessment**

**Comment 1:**

**This modeling and analysis are conducted in a satisfactory manner. However, it is hard to see how yet one more model that estimates evapotranspiration subcomponents moves us closer to a better understanding of these fluxes.**

**Reaction:**

Our aim is not to provide yet another LSM that partitions evaporation. Our aim is to show with a simple analytical model that the Budyko framework can be explained. For this we use the reasoning of the Gerrits model that recognizes the characteristic time scales of the different evaporation processes (i.e. interception daily and transpiration monthly). We revised the Gerrits model in such way that it was possible to apply it at the global scale. As suggested by the reviewer, we clarified this better and related the results of the model to the Budyko framework for a better understanding the partitioning of evaporation into transpiration and interception. For this paper, we changed the introduction for better explanation of our aim. Moreover, we provided Figures 7, 8 and 9 in the revised manuscript for better understanding of the evaporation partitioning through the Budyko framework.

**Comment 2:**

**The introduction and a paragraph in the discussion relate this model to the Budyko framework. One possible way forward for the authors is evaluating how trends in flux components relate the energy and water limitations outlined by the Budyko framework, since this is the stated motivation of this model. This could move the paper beyond how it is currently presented as another land surface model applied using remote sensing observations. For example, see Figure 11 of Miralles's 2016 HESS paper for casting total evaporative fluxes in this context. Also relevant is the study of Good (Nature Ecology & Evolution 2018) which used a Budyko approach to examine how to partition evaporative fluxes. In revising the paper, I suggest the authors work to find how this approach helps us understand the different surface to atmosphere water flux pathways better.**

**Reaction:**

We thank the reviewer for this valuable suggestion. We agree that our aim was not clearly defined and also misleading in that sense. As suggested by the reviewer, we evaluated the relation between evaporation fluxes and energy/water limitation in Budyko framework as provided by Miralles et al. (2016) and Good et al. (2017). As mentioned in reaction to comment 1, we provided Figures 7, 8 and 9 in the revised manuscript for each land cover, and for the evaporation fluxes ($E_i$ and $E_t$ and $E_{tot}$), separately, to discuss how our model can be related to the Budyko framework.

**Comment 3:**

**Most critically, I find the language in this paper to be grandiose and predicated on a poorly based argument. As is written in the abstract and introduction, the authors suggest that others have "tried to improve the Budyko framework by including more physics and catchment characteristics… However this often resulted in additional parameters, which are unknown or difficult to determine." This statement, and others like it in this paper, is inappropriate for two reasons: (1) other approaches have used fairly easy to measure characteristics and (2) because the authors proceed to do exactly what they claim shouldn't be done by fitting "difficult" to determine parameters to optimize their results. For point (1) for instance, the approach of Porporato is explicitly physically based as is it dependent on the ratio soil water storage to mean rainfall depth which is a measurable quantity. Furthermore, both of these quantities are used in the analysis presented here. For reason (2), the 'b' parameter of this analysis, among others, is clearly stated by the authors (P5L15) to have been calibrated to produce the best results. This is very similar to the Li (WRR 2013) paper wherein the Budyko curve parameters were fit to vegetation cover. The authors use of language such as "tried" (P2L18) seems to imply these other authors were unsuccessful, which may not be true. In my opinion, this submitted paper is quite similar to these other efforts in that it has extended the Budyko framework with new parameters they have fitted based on physical processes. Here, the most important parameters dictating the transpiration component are when transpiration becomes downregulated, and how much maximal transpiration can be. Equation 17 needs more elaboration and justification, as does the parameterization of Sb as 50% of S_u,max. How were these values selected and what is the consequence of other using other values here. How much do these choices, and other values such as the 'b' parameter, influence model outcomes.**

**Reaction:**

Yes, you are right that we also have some calibration parameters. Thus, we rephrased our text. Nonetheless, we think that we use a slightly different approach for these calibration parameters and other model parameters as well. Although others indeed also use 'measurable parameters', which could be tested in some case studies, some of these input values are not available at the global scale such as for example the soil water storage. For example, carry over parameter ($A$) was available for 10 locations in Gerrits et al. (2009), but at global scale we did not have such data, so we proposed $A=b*S_{u,max}$, and we need to calibrate the "$b$" parameter to link $A$ to a measurable variable. About the $S_b$ as 50% of $S_{u,max}$, we mentioned in the text that in this study we assumed $S_b$ to be 50% of $S_{u,max}$, as this value is commonly used for many crops, referred to (Allen et al. 1998). However, we provided a sensitivity analysis in the revised which shows that the model is not sensitive to this parameter for none of the land covers.

**Specific Comments:**

**Comment 4:**

**P1L11: The 1/(1+f(phi)) is not the base of all Budyko curves. Budyko, himself used a hyperbolic tangent as an example. What do the lower and upper case f's represent?**

**Reaction:**

As mentioned by Arora (2002), evaporation ratio ($E/P$) is a function of the aridity index ($\Phi$) and Bowen ratio ($\gamma$) ($\frac{E}{P} = \frac{\emptyset}{1+\gamma}$). Arora interpreted the equation as follows:

 "As a region becomes dry and is characterized by high potential evaporation, low precipitation and evapotranspiration, and high sensible heat fluxes then $\Phi\rightarrow1$, $\gamma\rightarrow1$ and $E/P$ tends towards unity implying little runoff. the other hand, as a region becomes wet and is characterized by low values of aridity index ($\Phi$) and Bowen ratio ($\gamma$) then $E/P < 1$ and runoff occurs. Since Bowen ratio ($\gamma$) is also a function of available energy and precipitation (and thus a function of $\Phi$) evaporation ratio may be expressed as a function of aridity index alone." It leads to equation 1 in our paper. Thus, in equation 1, f and F are both mathematical functions, showing that $E/P$ is a function of the aridity ($\Phi$). F($\Phi$) can have many forms (exponential, hyperbolic tangent, etc.) as summarized in Table 1.

**Comment 5:**

**P2L33: This paper estimates available soil water capacity, not the actual soil water itself. Also, I wouldn't call these 'data' but modeled estimates.**

**Reaction:**
Gao et al. (2014) presented a new method where the available soil water is derived from time series of rainfall and potential evaporation, plus a long-term runoff coefficient. We agree that knowing soil moisture storage change is important for the Budyko framework, but we use a method whereby we work around it by using plant available water. The method of Gao et al. (2014) provides plant available water (which is often linked to soil water capacity). In our paper we used it as $S_{u,max}$. We rephrased it in the manuscript to explain it more preciously.

Moreover, "data" refers to rainfall, potential evaporation and runoff coefficient which is used by Gao et al. (2014) to estimate the available soil water. However, we changed "data" into "**input time series**".

**Comment 6:**

**P3L16: Evaporation from 'non-superficial' soil moisture**

**Reaction:**

Thanks. We added this.

**Comment 7:**

**P4L11: Do you mean daily, not yearly, average.**

**Reaction:**

Yes, daily average during the year. We corrected it in the text.

**Comment 8:**

**P5L14: I think you should also place these eqn in table 2 for consistency: A = b\*S_u,max as well as Sb = 0.5\*S_u,max**

**Reaction:**

Ok, these equations moved to table 2 (equations 8 and 18, in the revised manuscript).

**Comment 9:**

**P5L36: Reword here. As is stated above and in eq17, you do not hold Dt,m constant? Which is it?**

**Reaction:**

We keep $D_{t,m}$ constant during the year (like $D_{i,d}$), but equation 17 shows that we calculated it as a function of the average yearly LAI. For water-constrained areas this is not a problem, because there $E_{t,m}$ is determined by the LHS of the min-function $(A + B(P_m - E_{i,m}))$ as can be seen in Equation 7. For energy-constrained areas our assumption can be problematic. However, in those areas often temperature and radiation follow a sinusoidal pattern without complex double seasonality as e.g., occurs in the ITCZ. This implies that the overestimation of $E_{t,m}$ in winter will be compensated (on the annual time scale) by the underestimation in summer time.

**Comment 10:**

**P5L38: Do you have a justification or citation for this statement?**

**Reaction:**

Please see our response to comment 9.

**Comment 11:**

**P7L17: No observations where used here. Only comparisons of the Gerrits model against other models.**

**Reaction:**

It is a general sentence for Taylor diagram, not only for our model. But to make it clear, we changed "This diagram can provide a concise..." into 'A Taylor diagram can provide a concise..."

**Comment 12:**

**P8L42: There are many bare soil estimates (See the review by Kool 2014 Agg and Forest Met, for example).**

**Reaction:**

We meant that there is hardly any data on the forest floor interception storage capacity ($S_{max}$). We did not intend to refer to bare soil evaporation.

**Comment 13:**

**F2: Because of the size of these figures, and the large range of values, it becomes hard to discern differences. Why not plot the absolute value of E flux in panel A, and then the differences in panels B, C, and D. Consider this approach in later figures as well**

**Reaction:**

Thanks for your suggestion. We did it before, but the single pixel outliers may blow up the entire figure what was also not a good way of showing the differences. That's why we moved towards the Taylor diagrams. Moreover, we also wanted to show the original data.

**Comment 14:**

**F3: Units for the RMSE, here and onward.**

**Reaction:**

We used normalized RMSE in these figures as shown in the following equation, so it has no unit.

$$\text{NRMSE} = \frac{\text{RMSE}}{\overline{X_O}} \qquad (1)$$

In this equation, NRMSE is normalized RMSE and $\overline{X_O}$ is the average of observation values (here the values estimated by Gerrits' model).

**Reviewer 2:**

**Comment 1:**

**Equation (3b): $E = E_i + E_t$. Since $E_i$ includes soil evaporation, I would suggest to interpret this as $ET = E + T$ where E is evaporation and T is transpiration.**

**Reaction:**
As suggested by Savenije (2004) and based on the definition of total evaporation provided by Shuttleworth (1993), we call the sum of interception ($E_i$), soil evaporation ($E_s$), transpiration ($E_t$), and evaporation from water bodies ($E_o$) as "evaporation" ($E$). Thus, we did not apply the term of "evapotranspiration" (ET), because we agree that they "are very different in terms of time scale, time of occurrence, physical characteristics, climatic feedback and isotope composition" as explained in Savenije (2004).

**Comment 2:**

**Page 3 Line 21: Does Et have the same definition as the Es defined in Line 16?**

**Reaction:**
Yes, as mentioned in reaction to comment 1, $E_t$ is transpiration which is evaporation from the soil moisture connected to the root zone, because that's where the trees get their water from.

**Comment 3:**

**Page 4 Lines 28-30: whys accounting for n is rarely necessary? Maybe it is better to explain it briefly here.**

**Reaction:**
Miralles et al. (2010) and Pearce and Rowe (1981) both mentioned that accounting for *n* is rarely necessary. Pearce and Rowe (1981) mentioned that "In many climates, however, such adjustments will not be necessary, or small enough that they can be neglected". In our interpretation this is because the number or times the interception storage can be filled and completely emptied is limited once we assume a drying time of ca. 4 hours, which is common. For 12 hours of day light, it means that *n* can be maximal 3 times. However, the chance that you have 4 storms every 4 hours, with a drying period of 4 hours, is rather small for most climates. We added this explanation in the revised manuscript.

**Comment 4:**

**Page 5 Line 36: If the inter-annual variability of the Dt,m has any impact on the results?**

**Reaction:**
We explained in the manuscript (page 5, line 36-38) that taking a constant value for $D_{t,m}$ can be problematic in energy-constrained areas. For water-constrained areas this is not a problem, because there $E_{t,m}$ is determined by the LHS of the min-function $(A + B(P_m - E_{i,m}))$ as can be seen in Equation 7. For energy-constrained areas our assumption can be problematic. However, in those areas often temperature and radiation follow a sinusoidal pattern without complex double seasonality as e.g., occurs in the ITCZ. This implies that the overestimation of $E_{t,m}$ in winter will be compensated (on the annual time scale) by the underestimation in summer time. In addition, Gerrits et al. (2009) provided a sensitivity analysis on the effect of different $D_{t,m}$ on total evaporation. Their results showed that total evaporation is sensitive to $D_{t,m}$ only once the annual rainfall exceed ±1000 mm/y. However, a sensitivity analysis conducted for clarifying this issue for some parameters and variables (Figure 10 in the revised manuscript).

**Comment 5:**

**Page 5 Lines 37-37: "But in those relatively wet areas transpiration is underestimated in summer, but overestimated in winter, which will cancel out on the annual scale." Delete the first "But"?**

**Reaction:**
Thanks, it was done in the revised manuscript.

**Comment 6:**

**Page 7 Line 32: year-1**

**Reaction:**
Thanks, it was corrected in the revised manuscript.

**Comment 7:**

**Page 8 Lines 2-3: Is there any analysis in this study to demonstrate that the precipitation is the major factor that caused the different results from different models?**

**Reaction:**
By providing a sensitivity analysis in the revised manuscript, we showed how the model is sensitive to number of rain days and rain months. Moreover, the results of sensitivity analysis conducted by Gerrits et al. (2009) shows that the results are significantly sensitive to change in $n_{r,d}$. We revised the manuscript as follows:

"Different precipitation products applied in the models are likely the reason for the differences. As found by Gerrits et al. (2009), the sensitivity of the model to the number of rain days and rain months especially for the higher rates of precipitation can be a probable reason for poor performance of a model especially for the forests with the highest amount of precipitation. In Section "Sensitivity analysis" we will elaborate on the sensitivity of these parameters on the global scale."

**Comment 8:**

**Page 9 Lines 27-32: The global transpiration ratio estimated by Gerrits' model is larger than nearly all of the other studies listed, is there any reason?**

**Reaction:**
Our transpiration ratio estimate is indeed in the higher range compared to other models/studies, however, the transpiration ratio estimated by Miralles et al. (2011) is higher than our model (80% in comparison to 71%). Moreover, our estimation is close to that of Sutanto (2015) (69%) and Good et al. (2015) (65%). Coenders-Gerrits et al. (2014) also found that based on the model of Jasechko et al. (2013) transpiration ratio changes between 35% and 80%, which is in line with our current findings. We added this information in the revised manuscript.

**Comment 9:**

**Page 10 Lines 27-29: Since the constant value of 0.935 mm in Equation 10 could be underestimated for the forest floor interception, then what value is advised for the forest floor?**

**Reaction:**
Forest floor evaporation can be modeled for each region based on its characteristics (e.g., Wang-Erlandsson et al. (2014)). Or typical values on $S_{max}$ for the forest floor can be found in Table 22.1 of Gerrits and Savenije (2011). For example, in the UK for Pine (Pinus sylvestris), it is 0.6-1.7 mm (Walsh and Voigt, 1977), in Australia for Eucalyptus, it is 1.7 mm (Putuhena and Cordery, 1996) and in Luxembourg for Beech (Fagus sylvatica), it is 1-2.8 mm (Gerrits et al., 2010).

[revised manuscript text omitted]

---

## Author Response (AR2)

**Dear Editor and Reviewers,**

Thank you for your valuable input on our manuscript. As requested by Reviewer #2 we thoroughly checked the entire paper for typos and grammatical errors. Furthermore, we redefined the justification of our study. Currently, many Budyko studies improved its performance by adding more physics and catchment characteristics. Although these additions might increase its performance, it hampers the application of the models at the global scale, since the required parameters are difficult to obtain globally. Our aim is to test whether the revised version of Gerrits' model WRR 2009 can overcome this issue. The Gerrits' model is based on a simple evaporation model, and in this study we test whether some constant parameters of the 2009-model could be replaced by spatially variable values as derived from remotely sensed data. To verify its performance, we compare our revised Gerrits' model to some advanced models, i.e. GLEAM, STEAM, Landflux-EVAL. The changes are provided in the manuscript as follows.

[revised manuscript text omitted]